# LEVERAGING DRIVER FIELD-OF-VIEW FOR MULTIMODAL EGO-TRAJECTORY PREDICTION

**M. Eren Akbiyik**[1], **Nedko Savov**[2], **Danda Pani Paudel**[2], **Nikola Popovic**[1,2],
**Christian Vater**[3], **Otmar Hilliges**[1], **Luc Van Gool**[2] **& Xi Wang**[1,4]

[1] ETH Zürich      [2] INSAIT, Sofia University "St. Kliment Ohridski"
[3] University of Bern      [4] TU Munich

## ABSTRACT

Understanding drivers' decision-making is crucial for road safety. Although predicting the ego-vehicle's path is valuable for driver-assistance systems, existing methods mainly focus on external factors like other vehicles' motions, often neglecting the driver's attention and intent. To address this gap, we infer the ego-trajectory by integrating the driver's gaze and the surrounding scene. We introduce RouteFormer, a novel multimodal ego-trajectory prediction network combining GPS data, environmental context, and driver field-of-view—comprising first-person video and gaze fixations. We also present the Path Complexity Index (PCI), a new metric for trajectory complexity that enables a more nuanced evaluation of challenging scenarios. To tackle data scarcity and enhance diversity, we introduce GEM, a comprehensive dataset of urban driving scenarios enriched with synchronized driver field-of-view and gaze data. Extensive evaluations on GEM and DR(eye)VE demonstrate that RouteFormer significantly outperforms state-of-the-art methods, achieving notable improvements in prediction accuracy across diverse conditions. Ablation studies reveal that incorporating driver field-of-view data yields significantly better average displacement error, especially in challenging scenarios with high PCI scores, underscoring the importance of modeling driver attention. All data and code is available at meakbiyik.github.io/routeformer.

## 1 INTRODUCTION

Understanding the perception and decision-making process of drivers is crucial for road safety in autonomous and assisted driving. Statistics reveal that 42% of car-bicycle collisions result from driver inattention (Allenbach et al., 2021). In response, a driver-assistance system capable of integrating the driver's perspective into its decision-making would enable vehicles to make informed decisions (Schwarting et al., 2018). Although predicting the driver's path is critical for such systems, traditional methods focus mainly on inferring the paths of other vehicles (Salzmann et al., 2020; Gu et al., 2021; Ngiam et al., 2021; Varadarajan et al., 2022; Nayakanti et al., 2023). In contrast, we seek to predict the drivers' ego-trajectory by observing both the surrounding scene, and the perception of the ego-vehicle's drivers via their frontal view and gaze fixations.

The central idea of this work builds on the insight that the perception of a person is tightly intertwined with their imminent and long-term goals, influenced by the surrounding traffic (Triesch et al., 2003; Hayhoe et al., 2003; Argyle et al., 1994). In driving, the attention of a driver can reveal otherwise inaccessible information about their decisions on the road. Complementing scene information with these cues enables the vehicles to resolve ambiguous situations. For instance, the head movements of the driver might be highly informative in determining the turn direction in a junction. The value of predicting ego-trajectories emerge in ambiguous, complex road paths, e.g., cases where the road is not trivially straight, or when predicting where the driver would turn at an intersection.

Our focus is thus twofold: firstly, to develop a framework capable of predicting the drivers' future ego-trajectories in various driving scenarios, and secondly, to create a method for identifying complex yet rare trajectories and quantifying their complexity. Despite the presence of other driving datasets with gaze data (Gopinath et al., 2021; Xia et al., 2017; Fang et al., 2023), to the best of

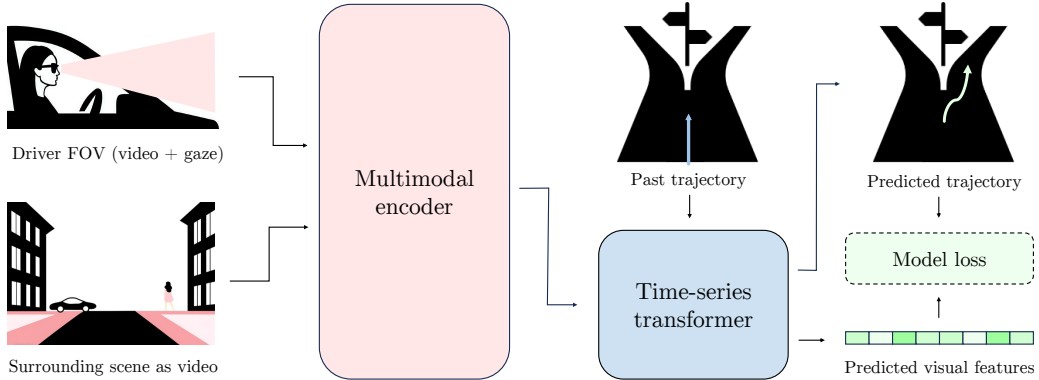

Figure 1: **RouteFormer framework**. Using the past GPS and the scene together with driver field-of-view, we predict the future ego-trajectory and visual features in driving with a novel loss scheme.

our knowledge, only one public benchmark DR(eye)VE contains both ego-drivers' locations and their field-of-view (Palazzi et al., 2018). The scarcity of available data poses another challenge for our task. To this end, we present an end-to-end framework that predicts future ego-trajectories, a novel metric to measure the complexity of road scenarios, and a new dataset of real-world driving recordings that include drivers' field-of-view changes.

First, we build a multi-modal framework, RouteFormer, for egocentric trajectory estimation that integrates scene data, driver FOV, and past trajectory, as shown in Figure 1. RouteFormer is crafted based on insights from time-series forecasting literature, offering the flexibility to incorporate additional data from other modalities, such as navigation paths and LIDAR. We train RouteFormer with auxiliary tasks of predicting future gaze and scene features for regularization, using a novel future-discounted loss formulation. To better quantify the complexity of various driving scenarios, we propose a new metric named *Path Complexity Index* (PCI), which measures the divergence of a driving trajectory from an extrapolation of the current path. It also indicates the difficulty of the corresponding prediction tasks and allows us to analyze model performance under different scenarios. Finally, we introduce a *Gaze-assisted Ego Motion* (GEM) dataset that captures diverse urban driving scenarios across 10 subjects, integrating GPS, high-resolution scene data from two front-view cameras, and driver field-of-view (FOV)—comprising first-person video and gaze fixations.

Experimental results show that RouteFormer outperforms state-of-the-art (SOTA) methods on the GEM dataset. To test its generalization capabilities, we also evaluate RouteFormer on the DR(eye)VE dataset (Palazzi et al., 2018), and our method surpasses the SOTA approaches. Ablation studies show that incorporating drivers' field-of-view information improves the prediction quality, particularly in high-PCI situations.

In summary, our contributions are:

- RouteFormer, an end-to-end multimodal ego-motion prediction network that effectively utilizes FOV data with a novel loss design.
- PCI, a metric that measures the trajectory complexity.
- GEM, an ego-motion dataset with driver positions and perspective.

## 2 RELATED WORK

**Ego-trajectory prediction.** Predicting the future sequence of locations of a moving entity has been explored for humans (Lyu et al., 2022; Rodin et al., 2021; Singh et al., 2016; Park et al., 2016), road agents (Huang et al., 2022; Gulzar et al., 2021) and aircraft (Zeng et al., 2022). Autonomous driving has largely focused on estimating the trajectories of the many agents surrounding the ego-vehicle (Salzmann et al., 2020; Gu et al., 2021; Nayakanti et al., 2023; Varadarajan et al., 2022; Ngiam et al., 2021). Recent advances in multi-agent prediction leverage specialized modalities like HD maps (Shi et al., 2022a) and interaction graphs (Girgis et al., 2021; Vishnu et al., 2023), while emerging approaches exploit large language models (Mao et al., 2023; Zheng et al., 2024) and human-inspired perception (Liao et al., 2024a;b). The fusion of these diverse modalities has shown

promise in improving prediction accuracy (Choi et al., 2021; Li et al., 2024a). However, predicting the ego-vehicles's trajectory, which is crucial for driving assistance systems (Schwarting et al., 2018; Jain et al., 2015; Vellenga et al., 2024; Kung et al., 2024), remains less explored. Kinetic models assuming constant velocity and turn rate have been utilized (Ammoun & Nashashibi, 2009). Kim et al. (2021) predict the ego-vehicle's trajectory with a VAE by conditioning on predicted driving style, while Baumann et al. (2018) use observations of the static environment, albeit neither with visual information, similarly with Kim et al. (2017) and Feng et al. (2019). Only Malla et al. (2020) proposes a vision module but requires hand-annotated actions. Taking a different direction, we focus on leveraging drivers' gaze and the scene to predict future ego-vehicle locations.

**Driving and attention.** Individuals tend to focus on relevant objects in situations with a particular goal in view, acquiring necessary information even before they are needed (Land & Lee, 1994; Underwood et al., 1999; Zhang et al., 2022; Fathi et al., 2012; Admoni & Srinivasa, 2016; Argyle et al., 1994). In navigation tasks, individuals often fixate on an intended path or destination well before initiating the movement (Hayhoe et al., 2003). For example, Zheng et al. (2022) successfully shows that the gaze is a high-quality indicator for human motion with GIMO. Similarly, numerous studies have also employed eye tracking to study the driving behavior of individuals, focusing on aspects such as attention (Ahlström et al., 2021), cognitive load (Engström et al., 2005; Kountouriotis et al., 2015), and levels of fatigue (Heitmann et al., 2001; Gao et al., 2015), while many of them are conducted in a driving simulator. For driver maneuver classification, recent work has shown relative success with an in-cabin camera observing the driver (Jain et al., 2015; Ma et al., 2023; Vellenga et al., 2024; Li et al., 2024b), with some works exploring using gaze and scene information for ego-action prediction, e.g. turns or lane changes (Lee et al., 2021; Amadori et al., 2020; Wu et al., 2019; Yi et al., 2023). Han et al. (2023) predict maneuvers based on rider sensor head motion data. Yan et al. (2023) use head motion and headset gaze to forecast the driver's future path by fitting a polynomial, but their non-robust gaze data limits its effectiveness. Finally, Ma et al. (2023) propose a transformer model, which integrates driver behavior information for maneuver prediction. In contrast, we use driver's field of view together with the surrounding scenes for the trajectory prediction task, inspired by similar works in human motion such as GIMO (Zheng et al., 2022).

**Driving datasets with driver FOV.** In the context of driving, there are datasets that provide in-vehicle footage of the driver's behavior as an indication of driver's attention (Ortega et al., 2020), zone-binned gaze (Ghosh et al., 2021; Ribeiro & Costa, 2019) (stationary), head pose (Schwarz et al., 2017). Jain et al. (2015) provide synchronized footage of the driver and the surrounding scene, yet without gaze data. Palazzi et al. (2018) provide a dataset of driving under various conditions, with gaze data from a tracking headset and synchronized GPS locations, however, with limited urban scenes. Given the limited amount of driving datasets with gaze, we introduce a new high-resolution dataset, where accurate eye tracker gaze location and vehicle GPS information are available. In contrast to others, we focus on city scenes with many agents where complex situations can arise.

## 3 EGO-MOTION PREDICTION WITH DRIVER FIELD-OF-VIEW

Driving is an interplay of short and long-term goals, influenced by exogenous factors. Although such external effects can be understood to some level through computer vision, drivers' behavior is mostly driven by self-established targets, which complicates the task of ego-motion prediction. To address this challenge, we propose a new framework named RouteFormer. At its core, our method is designed to exploit visual information from multiple cameras, the past trajectory of the vehicle, and, most notably, the driver's field-of-view—comprising first-person video and gaze fixations. Each modality contributes to understanding a different aspect: the surrounding scene, the motion of the vehicle, and the targets of the driver. See Figure 2 for an overview.

RouteFormer offers two key contributions: (1) a novel architecture that fuses multimodal inputs via self- and cross-attention mechanisms for time-series prediction, and (2) an auxiliary loss and future-discounting to regularize long-term forecasting that is otherwise prone to over-fitting.

### 3.1 TASK DEFINITION

In the ego-trajectory prediction task, given measurements of the vehicle and the environment for $T$ time steps, we aim to anticipate the future locations of the vehicle for each of the next $T_{\text{pred}}$ time

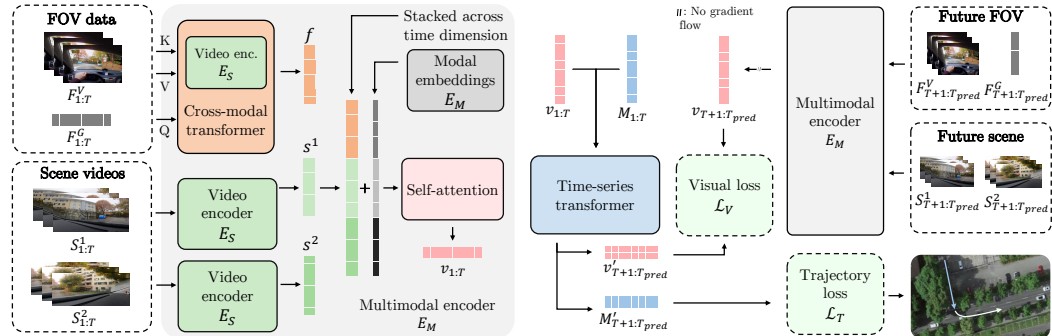

(a) Multimodal encoder for FOV and scenes     (b) Time-series module and loss configuration

Figure 2: **Framework details.** The multimodal architecture fuses FOV, scene, and motion data for ego-trajectory forecasting. (a) The videos are encoded with the scene encoder $E_S$ frame-wise using a pre-trained vision backbone. FOV data is then encoded via a cross-modal transformer. The resulting tensors, all in the image feature domain, are stacked across time, self-attended, and concatenated with motion features for forecasting. (b) RouteFormer predicts the trajectory, as well as features from visual modalities concurrently to use them as auxiliary losses for regularization.

steps (our prediction horizon). For the first $T$ steps, we use as input the vehicle's motion $M_{1:T}$ and scene representations in the form of video from K cameras of the surroundings $S_{1:T}^{1:K}$. Notably, in addition, we incorporate FOV data $F_{1:T}$. We build a model $\Psi(M_{1:T}, S_{1:T}^{1:K}, F_{1:T}|\lambda) = M'_{T+1:T_{\text{pred}}}$ to predict the future motion, where $\lambda$ are the model's parameters.

We represent vehicle location with two GPS coordinates using the EPSG:3857 coordinate system (in meters) (Pridal & Pohanka, 2024). The $t$'th element of the input motion $M_{1:T} \in \mathbb{R}^{T \times 2}$ corresponds to the relative change of the coordinates compared to $t - 1$, as in Salzmann et al. (2021). For each time step $t \in [0, T]$, the scene $S_{1:T}^{1:K} \in \mathbb{R}^{K \times T \times 3 \times H_S^K \times W_S^K}$ contains images from K external vehicle cameras, with width $W_S^K$ and height $H_S^K$. FOV data has two components $F_{1:T} = \{F_{1:T}^V; F_{1:T}^L\}$ representing a frontal video feed from a camera fitted to the driver's head $F_{1:T}^V \in \mathbb{R}^{T \times 3 \times H_F \times W_F}$, and the high-frequency gaze positions, respectively. The second component, $F_{1:T}^L \in \mathbb{R}^{T \times 2}$, consists of $(x, y)$ coordinates of points gazed in the image $F^V$, relative to the bottom-left of the image.

## 3.2 ROUTEFORMER ARCHITECTURE

In our architecture, scene and FOV modalities are processed by separate encoders $E_S(S_{1:T}^{1:K}) = s_{1:T}^{1:K}$ and $E_F(F_{1:T}) = f_{1:T}$, producing sequences of compact feature vectors of the same size. Next, all K scene features are fused with FOV features using a self-attention module $E_V(s_{1:T}^{1:K}, f_{1:T}) = v \in \mathbb{R}^{T \times D}$. Finally, a time-series forecasting module $\mathcal{T}$ predicts the future $T_{pred}$ data points - the future motion, in addition to feature vectors of other modalities used in training. We design two versions of this architecture: RouteFormer-Base, $\Psi(M_{1:T}, S_{1:T}^{1:K}|\lambda)$, which does not use the FOV features, and RouteFormer, $\Psi(M_{1:T}, S_{1:T}^{1:K}, F_{1:T}|\lambda)$, which incorporates all the available modalities, including the driver's perspective.

**Scene Module.** Our shared scene module $E_S$ encodes each frame in $S_{1:T}^{1:K}$ using a pre-trained vision backbone, in our case SwinV2 (Liu et al., 2021). The features are then reduced through a self-attention layer to produce $s_{1:T} \in \mathbb{R}^{T \times D}$ (see Figure 2 a). These features are used in subsequent layers for temporal reasoning with inter-frame attention.

**FOV Module.** The frames $F_{1:T}^V$ of the FOV data $F_{1:T}$ are encoded using the same scene encoder $E_S$, in order to share the representation space and allow for implicit registration between the images. To account for high frame rate of gaze positions and refine the potentially noisy gaze positions, we perform a multi-head self-attention of $F_{1:T}^G$, following the decoder design in Zhou et al. (2021). After achieving two encoded feature vectors of the same shape for gaze and frontal video, we perform gaze-frame cross-attention between visual and gaze location features. The attention mechanism allows the model to "pick" the relevant scene features using the gaze positions across frames, thus accounting for the highly dynamic scene in a moving vehicle. The vector produced is $f_{1:T} \in \mathbb{R}^{T \times D}$.

**Modality Fusion.** The next step is using an encoder $E_V$ to fuse scene $s_{1:T}^{1:K}$ and optionally FOV $f_{1:T}$ features. To this end, we concatenate $s_{1:T}^{1:K}$ and $f_{1:T}$ across the time dimension, sum with the modality embeddings $E_M$, and apply a self-attention mechanism. $E_M$ encodes the feature source (e.g. one of the scenes ($k \in [0, K]$) or FOV). This self-attention layer performs implicit image registration and feature selection within and between the scene and FOV modalities. More precisely, we take the concatenation $[s_{1:T}^{1:K}, f_{1:T}] \in \mathbb{R}^{(K+1) \cdot T \times D}$ and produce its fused encoding $v \in \mathbb{R}^{T \times D}$. In the case of RouteFormer-Base, where gaze data is not available, we only fuse the scene vectors.

**Time-Series Forecasting.** Next, we merge (along the feature dimension) the motion features $m_{1:T}$ and the aforementioned fused features $v_{1:T}$ to obtain $[m_{1:T}, v_{1:T}] \in \mathbb{R}^{T \times 2 \cdot D}$. This forms the input for our time-series forecasting component $\mathcal{T}$, which takes the merged features $[m_{1:T}, v_{1:T}]$ and produces a prediction of the future motion $M'_{T:T_{\text{pred}}}$. For this task, we experiment with the state-of-the-art long-term time-series prediction methods Informer (Zhou et al., 2021), Transformer (Vaswani et al., 2017), PatchTST (Nie et al., 2023) and NLinear/DLinear (Zeng et al., 2023). While the focus is on future trajectory prediction, we also predict future visual features $v'_{T:T_{\text{pred}}}$. The auxiliary outputs are essential for regularization during training by enforcing intermediate features to remain rich in scene- and FOV-related information, whenever available.

## 3.3 Training Objectives

Trajectory prediction models are prone to over-fitting due to the inherent difficulty of the training regime: as the subsequent predictions depend on the correctness of the previous forecasts, models might learn from spurious correlations of the later sections of the trajectory. To tackle this problem, RouteFormer employs a novel loss formulation which we term as "future-discounted loss". The principle behind this loss is to consider predictions for future time steps with diminishing weight, particularly at the early stages of the training, making predictions for the immediate future more influential on the overall loss than those for the distant future. This also follows real-world use cases where immediate predictions often have a higher impact than long-term predictions.

**Future-discounted loss.** Given a predicted sequence $\mathbf{y}_{pred}$ and a ground truth $\mathbf{y}_{gt}$, the future-discounted loss $\mathcal{L}_{\text{fd}}$ over $N$ time steps is then defined as:

$$\mathcal{L}_{\text{fd}}(\mathbf{y}_{pred}, \mathbf{y}_{gt}) = \sum_{i=1}^{N} \gamma^i (\mathbf{y}_{pred,i} - \mathbf{y}_{gt,i})^2$$

where $\gamma$ is the discount factor with $0 < \gamma < 1$. This is inspired by reward discounting in reinforcement learning literature, where the discount rate is often in $[0.9, 0.99]$. On 6s at 5fps, the choice of 0.97, as used in our experiments, gives $0.97^{30} = 0.4$ weight in the final time step.

**Loss composition.** Both the primary trajectory loss and the auxiliary losses are subject to the future-discounting. Let $\mathcal{L}_T$ denote the discounted trajectory loss and $\mathcal{L}_V$ the discounted video visual. See Appendix A.1 for detailed definitions. The combined loss $\mathcal{L}_{\text{combined}}$ can be expressed as:

$$\mathcal{L}_{\text{combined}} = \mathcal{L}_T + \alpha_V \mathcal{L}_V.$$

The coefficient $\alpha_V$ dynamically balances the primary and auxiliary losses. It is determined by:

$$\alpha_V = \rho_V \times \frac{||\mathcal{L}_T||}{||\mathcal{L}_V||}.$$

Here, $\rho_V$ is a scalar hyperparameter and $|| \cdot ||$ represents the parameter's scalar value detached from gradients. This ensures auxiliary losses maintain a consistent proportion to $\mathcal{L}_T$, stabilizing training while preserving trajectory prediction as the primary objective.

## 3.4 Implementation Details

RouteFormer is trained using the AdamW optimizer with a linear warm-up of 2 epochs and cosine annealing, over a total of 200 epochs. Maximum learning rate of $1 \times 10^{-5}$ and weight decay of $1 \times 10^{-4}$ are used with batch size 16. A full set of hyperparameters can be found in Appendix

**Regularization.** We employ no dropout inside the transformers, but apply aggressive within-modality dropouts (e.g., dropping one of the two scene videos) to improve validation scores.

**Caching.** To benefit from pre-trained vision backbones while keeping the training duration under control, we have implemented module-level caching that utilizes intermediate features extracted from these backbones. Details of the caching mechanism are explained in Appendix

## 4 PATH COMPLEXITY INDEX

A critical challenge in autonomous driving is tackling the long tail of driving scenarios - situations that are less frequent but potentially high-impact. These scenarios often encompass a diverse range of unpredictable driving conditions, such as navigating complex intersections or road obstructions. Traditional approaches often view the task as a classification problem, constrained by the limitations of hand labeling such events (Teeti et al., 2022; Girase et al., 2021; Palazzi et al., 2018). To automatically identify and quantify the complexity of driving situations, we introduce a novel metric termed *Path Complexity Index* (PCI). This metric evaluates the deviation from simple driving patterns, thereby providing a robust tool for assessing and improving trajectory prediction models.

Given an input trajectory, the PCI metric computes the Fréchet distance between the target trajectory and a hypothetical simple trajectory in which the driver maintains the speed and direction exhibited in the last segments of the input trajectory. Formally, the PCI of an observed target trajectory $\mathcal{T}_{target} \in \mathbb{R}^{T \times 2}$ given an input trajectory $\mathcal{T}_{input} \in \mathbb{R}^{T' \times 2}$, is given by:

$$PCI(\mathcal{T}_{target}|\mathcal{T}_{input}) =$$

$$\inf_{\alpha,\beta \in [0,1]} \max \left( \sup_{t \in [0,1]} \|\mathcal{T}_{target}(\alpha(t)) - \mathcal{T}_{simple}(\beta(t))\| \right)$$

where $\alpha : [0,1] \rightarrow [0,1]$ and $\beta : [0,1] \rightarrow [0,1]$ are continuous non-decreasing functions that represent all path reparameterizations (Eiter & Mannila, 1994), and $\mathcal{T}_{simple} \in \mathbb{R}^{T \times 2}$ is the simple trajectory derived by following the final velocity vector of $\mathcal{T}_{input}$:

$$\mathcal{T}_{simple}(t) = \mathcal{T}_{input}(T) + v_{final} \cdot t, \quad t \geq T$$

Here, $v_{final} = \mathcal{T}_{input}(T') - \mathcal{T}_{input}(T'-1)$ represents the velocity vector estimated from the final two points of $\mathcal{T}_{input}$, and $t$ is the time parameter extending beyond the duration of the input trajectory $T'$. $\|\cdot\|$ denotes the Euclidean distance in the 2D plane.

A high value of PCI indicates a significant deviation from the simple trajectory. Some generated examples of deviations from a straight input trajectory and their respective PCI range can be seen in Figure 3. A sample of real trajectories for these ranges are provided in Figure 4. We propose that a high PCI indicates intriguing events, which are often characterized as sudden changes in direction or speed in the literature (Girase et al., 2021). We provide more insights into this behavior in Appendix with a comprehensive breakdown of the PCI statistics in GEM dataset.

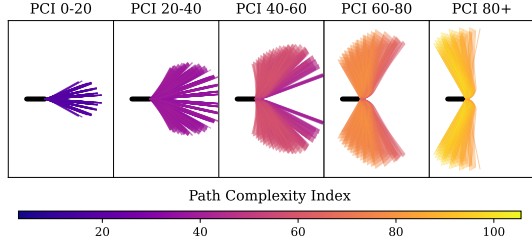

Figure 3: **Generated trajectories and their values**. The black paths to the left are inputs, and the colored paths are targets generated exhaustively by varying the speed, turning angle, and turn curvature.

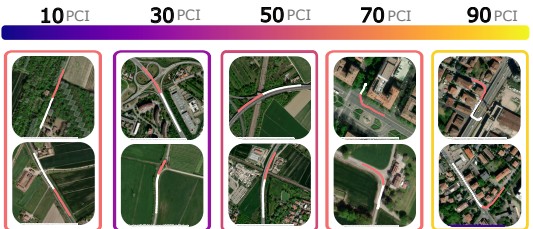

Figure 4: **Example trajectories with varying PCI**. White is the input and red is the target trajectory.

**Are all curved paths interesting?** An essential task for trajectory prediction models is learning the road curvature from scene information. PCI is simplistically designed to not bias towards a particular type of curvature to encourage that while allowing us to eliminate instances of straight cruising that may dominate the training and evaluation phases.

Table 1: **Contribution of GEM.** Our dataset is designed for ego-trajectory prediction with noise-free GPS positions and gaze data, in an entity-rich urban setting.

(a) **Comparison of existing trajectory prediction datasets with ours.** Only public datasets are considered. *: hand-annotated.

| Dataset | Gaze | Noise-free pose | Scene cameras | Duration |
|---|---|---|---|---|
| Waymo (Ettinger et al., 2021) | ✗ | ✓ | 5 | 574h |
| nuScenes (Caesar et al., 2020) | ✗ | ✓ | 6 | 5.5h |
| Argoverse (Wilson et al., 2021) | ✗ | ✓* | 7 | 305h |
| LOKI (Girase et al., 2021) | ✗ | ✗ | 1 | 2.3h |
| DR(eye)VE (Palazzi et al., 2018) | ✓ | ✗ | 1 (+1 headcam) | 6h |
| **GEM (Ours)** | ✓ | ✓* | 2 (+1 headcam) | 5h |

(b) **GEM vs. DR(eye)VE**: percentage of frames with other agents, and total entity counts, both detected by YOLOV8 at 1 fps.

| | GEM | DR(eye)VE |
|---|---|---|
| % of frames with other drivers | **89.7%** | 38.1% |
| % of frames with pedestrians | **31.2%** | 22.5% |
| # of cars | **23331** | 3979 |
| # of pedestrians | **3640** | 297 |
| # of buses | **1215** | 155 |
| # of bicycles | **357** | 18 |

## 5 GAZE-ASSISTED EGO MOTION (GEM) DATASET

The driver's gaze plays a significant role during driving, providing insights about their decisions on the road. Understanding this role may allow us to build systems to improve vehicular safety. However, training models in the gaze domain is constrained by data scarcity. The sole public dataset featuring ego-driver positions and gaze, DR(eye)VE, contains only 6 hours of data, only one-third of which is urban driving, hindering the capture of rare gaze-based cues. (Palazzi et al., 2018).

We, therefore, propose GEM, a multimodal ego-motion with gaze dataset with synchronized multi-cam video footage of the road scene and FOV data, with precise gaze locations from an eye-tracking headset, and GPS ego-vehicle locations, manually corrected for further accuracy. By coupling gaze data with the car's motion and the driving scene, our dataset provides a rich source of information for ego-trajectory prediction and modeling of drivers' behavior. The primary objective of our dataset is to enable the development and validation of models that can predict future ego-vehicle locations based on the driver head/eye movements and surrounding scenes.

### 5.1 HARDWARE SETUP

We mount high-resolution cameras on the vehicle to ensure high-quality recordings of the surrounding environment and record drivers' eye gaze using a head-mounted eye tracker.

**Frontal cameras and GPS tracking.** To record the driving scene and car motion, we use two GoPro cameras with a 4K resolution and a 30Hz sampling rate. These cameras record the car's GPS positions in real-time and embed this information into the video streams. However, GPS data can have discrepancies due to various factors such as satellite interference or obstructions, which we further correct.

**Gaze tracking.** We employ the Pupil Invisible glasses [1] from Pupil Labs to track the driver's gaze. This device captures eye movements at 200Hz using two near-eye infrared cameras and includes a frontal video camera of the scene. The gaze tracker and GoPro cameras are fully synchronized in post-processing using the timestamp metadata.

### 5.2 DATA COLLECTION

10 participants were instructed to drive naturally on a variety of road types with a focus on urban areas, during different times of the day to capture a comprehensive range of driving conditions. Each participant drove for approximately 30 minutes, resulting in a total of 5 hours of collected data. See Appendix D.1 for a breakdown of the dataset.

**Procedure.** Upon arrival, participants were briefed about the study and given a tutorial on the equipment. Before commencing the driving sessions, participants went through a gaze calibration process. They were asked to look at specific points on a board, ensuring that the eye-tracking device was accurately capturing their fixations. After fitting the Pupil Invisible glasses and ensuring calibration, participants were asked to start their driving session. A researcher was present in the car to oversee the equipment and address any unexpected situations.

---

[1]Tech Specs.

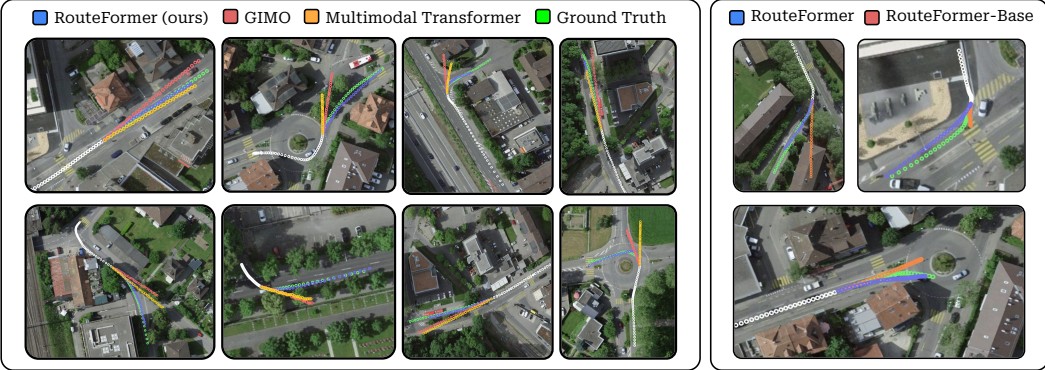

Figure 5: **Qualitative examples.** RouteFormer shows higher confidence in sharp turns than other SOTA models using gaze, which tend to prefer the mean of the past trajectory (left). The turn confidence is lower when no driver FOV information is used (right).

**Post-processing.** GPS technology inherently exhibits limitations, including potential positional inaccuracies up to 1.82m at the 95th percentile, as reported by the U.S. FAA (U.S. FAA NSTB/WAAS T&E Team, 2021). These inaccuracies are notably magnified in occluded environments, making the resulting data unsuitable for ground truth applications (see Appendix for examples). To mitigate this, we developed a user-operated application that allows for the manual correction of GPS markers to more accurate positions. This correction tool has been instrumental in enhancing the positional accuracy of all samples within the GEM dataset. The tool's source code will be released alongside the dataset. We provide the data split and detailed description of the dataset in the Appendix.

### 5.3 COMPARISON WITH EXISTING DATASETS

Most driving datasets to create road assistance systems lack either gaze data, noise-free vehicle trajectory, or sufficient quality scene information, as seen in Table 1 (a). To the best of the authors' knowledge, the only other public multimodal driving dataset with gaze data is DR(eye)VE, as presented by Palazzi et al. (2018). However, this dataset includes mostly rural/highway driving (63.5% of the data), while GEM is designed for complex urban trajectory prediction with many interacting agents. Our comparison in Table 1 (b), using YOLOV8 (Jocher et al., 2023), shows GEM's significant lead in terms of traffic-related content: it has other vehicles in 89.7% of frames (compared to DR(eye)VE's 38.1%), six times more cars, over ten times more pedestrians, and twenty times more bicycles than DR(eye)VE. For GEM, we further ensured the high quality of the GPS information, fixing the inherent GPS limitations through hand annotation, especially in occluded areas like tunnels (see in Appendix C for examples, and the dataset noise comparison).

## 6 EXPERIMENTS

We evaluate our method RouteFormer by measuring the predicted motion trajectories using a set of standard metrics. We compare against several state-of-the-art methods and baselines. See Appendix B for more results and examples.

### 6.1 SETUP

**Datasets.** We evaluate our method on our proposed GEM dataset. To demonstrate that the benefit from gaze extends beyond our dataset, we additionally train and evaluate our framework on a second dataset. **DR(eye)VE** (Palazzi et al., 2018) has 6 hours of driving along with gaze input, however, with a smaller number of traffic entities than GEM, and lower data quality.

**Task.** Following the task setting in nuScenes (Caesar et al., 2020), our task uses 8-second input trajectories to predict the next 6-second motion. We sample synchronized sequences of location, videos, and gaze data with a sliding window of 2-second strides, resulting in approximately 16 hours of input for the training split. We choose different drivers for training and evaluation according to the data splits of either dataset (see Appendix D) to better demonstrate the benefit of FOV.

Table 2: **Comparison to baselines and SOTA.** We use the test sets of both the proposed dataset and the DR(eye)VE dataset. Our model RouteFormer achieves the best performance on both datasets.

| Method | GEM | | DR(eye)VE | |
| --- | --- | --- | --- | --- |
| | ADE (m) | ADE+20PCI (m) | ADE (m) | ADE+20PCI (m) |
| Stationary Baseline | 32.04 | 30.22 | 64.11 | 61.41 |
| Linear Baseline | 7.37 | 13.37 | 10.16 | 15.54 |
| GIMO (Zheng et al., 2022) | 7.61 | 10.03 | 13.97 | 18.68 |
| Multimodal Transformer (Li et al., 2021) | 6.07 | 8.41 | 10.67 | 14.65 |
| Ours - RouteFormer-Base (video only) | 6.13 | 8.14 | 9.95 | 13.15 |
| Ours - RouteFormer (video+gaze) | **5.99** | **7.70** | **8.75** | **12.26** |

Table 3: **Varying the prediction horizon.** Average final displacement errors on GEM for predicting 1-6s in the future for test set samples with 20+ PCI, demonstrating RouteFormer's increasingly higher accuracy in long-term forecasting for complex samples, compared to RouteFormer-Base.

| Method | FDE@1s | FDE@2s | FDE@3s | FDE@4s | FDE@5s | FDE@6s |
| --- | --- | --- | --- | --- | --- | --- |
| RouteFormer-Base (video only) | 1.74 | 4.19 | 7.08 | 10.56 | 14.45 | 18.45 |
| RouteFormer (video+gaze) | 1.68 | 4.04 | 6.72 | 9.93 | 13.61 | 17.37 |
| Improvement | 3.57% | 3.71% | 5.36% | 6.34% | 6.17% | 6.21% |

**Baselines.** We use two simple baselines. **Stationary** baseline assumes that the vehicle remains stationary throughout the prediction window. **Linear** baseline assumes that the vehicle maintains a linear trajectory, following the vehicle's final direction and speed from the input.

**Motion prediction models.** We compare our model with two gaze-assisted human motion prediction models adapted from the literature. **Multimodal Transformer** (Li et al., 2021) encodes the input modalities with linear layers and predicts the future trajectory with an encoder/decoder architecture. In addition, we implement **GIMO** (Zheng et al., 2022) to utilize scene videos using RouteFormer's own video encoder instead of 3D point clouds and predict the future trajectory with a cross-modal encoder architecture. The adaptations follow Zheng et al. (2022)'s adjustments in GIMO to compare with their baseline Multimodal Transformer (referred to in Tab. 2).

**Metrics.** We use the standard evaluation metrics Average Displacement Error (ADE) and Final Displacement Error (FDE). **ADE** measures the average error between the ground truth and predicted trajectory across all time steps. **FDE** measures the displacement error in the final prediction time step, emphasizing the model's accuracy in predicting the endpoint of a trajectory.

## 6.2 EVALUATION OF ROUTEFORMER

**Comparison to baselines and SOTA.** We report results on the test sets of both our proposed dataset GEM and the DR(eye)VE dataset in Table 2. We train on 8 participants and test on 2 others to explore generalizability across people. Our model RouteFormer outperforms all baselines across both evaluation metrics and, therefore achieving state-of-the-art performance on both datasets.

**Qualitative evaluation.** In Figure 5, we display the behavior of the proposed model compared to SOTA and baselines. RouteFormer shows better confidence in sharper turns compared to other models, predicts right exits in roundabouts, and overall has better performance in high-PCI situations.

**Evaluating the benefit of driver FOV.** We compare RouteFormer with RouteFormer-Base, its version without driver FOV (Table 2). The additional modality in RouteFormer significantly improves performance, especially for later points in the prediction horizon (Table 3). This confirms the importance of driver field-of-view information in ego-trajectory prediction, particularly for long-term forecasts. Notably, RouteFormer achieves lane-level accuracy for shorter horizons even in challenging GEM samples, demonstrating potential for industrial short-term forecasting applications.

**Path complexity evaluation.** We evaluated RouteFormer and RouteFormer-Base across different levels of path complexity using the PCI metric. Results in Figure 6 show that gaze data becomes increasingly valuable for more irregular trajectories. This demonstrates RouteFormer's ability to generalize driver FOV learning, improving predictions for complex paths across different drivers.

**Gaze fixation importance.** Throughout this study, we consider driver's FOV as a composition of a first-person video feed and gaze fixations, as they are highly interconnected and novel remote tracking systems (such as Smart Eye Pro) can capture them together (Smart Eye AB, 2024). To verify the importance of gaze fixations beyond first-person video, we analyzed their attribution using Inte-

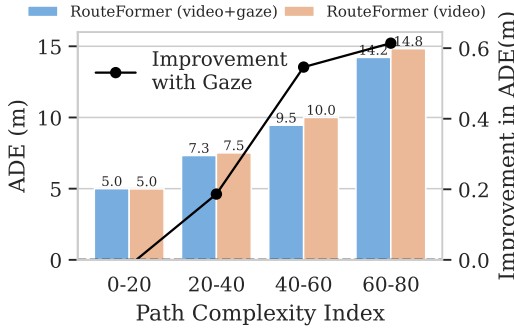 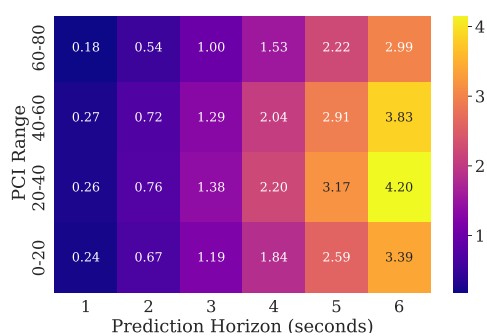

Figure 6: **The effect of gaze over different PCI.** Lower is better for ADE. We bin GEM's test set by PCI, displaying average ADE per group. The line represents the ADE difference between RouteFormer-Base and RouteFormer, with higher values favoring RouteFormer.

Figure 7: **Attribution to gaze fixations across different PCI bins and prediction horizons.** Gaze fixations increasingly affect the prediction for further away horizons and medium/high PCI values. Estimated using Integrated Gradients (Sundararajan et al., 2017).

grated Gradients (Figure 7). Fixations significantly impact the final result, with increased reliance for later prediction points and higher PCI samples. This confirms that RouteFormer effectively utilizes both video and gaze data, despite having the same parameter count as RouteFormer-Base.

## 6.3 ABLATION STUDIES

We performed a set of ablation studies to choose the right vision backbones and time series modules for RouteFormer, and justify the discounted/auxiliary losses.

**Auxiliary and discounted losses.** As one of the major contributions of our framework, we assess the effect of the future-discounted loss $\mathcal{L}_{fd}$ and the auxiliary losses. Both additions show significant improvement over the standard architecture on the validation set (Table 4).

**Time-series module.** Informer (Zhou et al., 2021) is a Transformer-based model and leverages the ProbSparse attention mechanism. Table 5 shows that Informer consistently outperforms later models such as Autoformer (Wu et al., 2021), LTSF-Linear (Zeng et al., 2023), and PatchTST (Nie et al., 2023). We used Informer in RouteFormer.

Table 4: **Loss ablations.** Results over GEM validation set for 20+ PCI samples.

| Method | Val. ADE (m) | Val. FDE (m) |
|---|---|---|
| Vanilla RouteFormer | 8.85 | 58.54 |
| + auxiliary losses | 8.68 (-0.17) | 57.09 (-1.45) |
| + future-discounted loss | 8.27 (-0.41) | 54.38 (-2.71) |

Table 5: **Visual encoder and time-series module ablations.** Results over GEM val. set for 20+ PCI samples. Time-series modules (top) perform prediction without scene or gaze data. Pretrained vision modules (bottom) use Informer.

| Method | Val. ADE (m) | Val. FDE (m) |
|---|---|---|
| DLinear (Zeng et al., 2023) | 10.67 | 81.40 |
| NLinear (Zeng et al., 2023) | 10.49 | 83.38 |
| PatchTST (Nie et al., 2023) | 10.23 | 79.80 |
| Transformer (Vaswani et al., 2017) | 9.97 | 67.88 |
| Informer (Zhou et al., 2021) | **9.62** | **66.14** |
| Informer+SAM (Kirillov et al., 2023) | 9.08 | 57.89 |
| Informer+DinoV2 (Oquab et al., 2023) | 8.69 | 60.83 |
| Informer+SwinV2 (Liu et al., 2021) | **8.51** | **55.56** |

**Pretrained vision backbone.** In Table 5, we evaluate the impact of various visual encoders, including SAM (Segment Anything) (Kirillov et al., 2023), DinoV2 (Oquab et al., 2023) and SwinV2 (Liu et al., 2021). SwinV2 consistently demonstrated superior performance on the validation set with embedding sizes smaller than its counterparts.

## 7 CONCLUSION

Our results highlight the significance of incorporating driver FOV in predicting ego-vehicle trajectories. Through our innovative multimodal method, RouteFormer, which integrates FOV, scene and motion data, we achieve superior performance over existing baselines, pushing the boundaries of state-of-the-art achievements in this domain. Specifically, our approach demonstrates that incorporating first-person video and gaze fixations data enhances the prediction of complex non-linear trajectories. By establishing a new benchmark with our GEM dataset, we pave the way for further intuitive and human-centric developments in assisted driving technologies.

ACKNOWLEDGEMENTS

INSAIT, Sofia University "St. Kliment Ohridski". Partially funded by the Ministry of Education and Science of Bulgaria's support for INSAIT as part of the Bulgarian National Roadmap for Research Infrastructure. This project was supported with computational resources provided by Google Cloud Platform (GCP). This project is also partially funded by the German Federal Ministry of Education and Research through the ExperTeam4KI funding program for UDance (Grant No. 01IS24064).

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

# APPENDIX

## A   ROUTEFORMER IMPLEMENTATION DETAILS

All experiments are carried out on a machine with an NVIDIA GPU with $> 10$ GB memory, 64 GB of RAM, and an Intel CPU. The framework is implemented using PyTorch 2.0.

### A.1   ROUTEFORMER NOTATION

Using the notation we have established, our full model can be described as:

$$\Psi\left(M_{1:T}, S_{1:T}^{1:K}, F_{1:T}|\lambda\right) \equiv$$
$$\mathcal{T}\left(M_{1:T}, E_P(E_S(S_{1:T}^{1:K}), E_F(F_{1:T}))\right) =$$
$$\{M'_{T:T_{\text{pred}}}, v'_{T:T_{\text{pred}}}\}$$

**Loss definitions.**   For the model outputs $\Psi\left(M_{1:T}, S_{1:T}^{1:K}, F_{1:T}|\lambda\right) = \{M'_{T:T_{\text{pred}}}, v'_{T:T_{\text{pred}}}\}$, let $\mathcal{L}_T$ denote the discounted trajectory loss, and $\mathcal{L}_V$ the discounted visual loss. The auxiliary losses are defined as:

$$\mathcal{L}_T = \mathcal{L}_{\text{fd}}\left(M'_{T:T_{\text{pred}}}, M_{T:T_{\text{pred}}}\right)$$
$$\mathcal{L}_V = \mathcal{L}_{\text{fd}}\left(v'_{T:T_{\text{pred}}}, E_P(E_S(S_{T:T_{\text{pred}}}^{1:K}), E_F(F_{T:T_{\text{pred}}}))\right).$$

## A.2 Module-Level Caching with TorchCache

Caching is a crucial component of our training regime. During the initial epochs, the intermediate representations from the pre-trained vision backbones are computed for each input frame and stored in an efficient cache memory structure. When subsequent epochs require the same input data, the pre-computed representations are fetched from the cache, thereby eliminating the need for repetitive and costly forward passes through the vision backbones.

To make this process repeatable, we implemented the TorchCache utility to provide a generic caching mechanism. The TorchCache library offers an easy way to cache the outputs of PyTorch modules, making it particularly suited for our need to efficiently cache the outputs of large, pre-trained vision transformers. The package of this mechanism is open-sourced at github.com/meakbiyik/torchcache.

### A.2.1 Key Features

The TorchCache library offers several notable features:

- Allows caching of PyTorch module outputs in two formats: in-memory for fast retrieval, and persistently on the disk for larger datasets.
- Adopts a decorator-based interface that simplifies its integration with PyTorch modules.
- Implements an MRU ( most-recently-used) cache policy to manage and limit memory or disk consumption.

### A.2.2 Usage

The library's decorator-based design ensures minimal modifications to the existing PyTorch codebase. By adding the `@torchcache()` decorator, the output of the module's forward method gets cached, as illustrated in the provided basic usage example:

```python
from torchcache import torchcache

@torchcache()
class MyModule(nn.Module):
    def init(self):
        super().init()
        self.linear = nn.Linear(10, 10)
    def forward(self, x):
        # This output will be cached
        return self.linear(x)
```

More detailed examples, advanced usage patterns, and further documentation will be accessible at the official TorchCache documentation.

### A.2.3 Operational Assumptions

For the TorchCache library to operate seamlessly, there are certain assumptions that the PyTorch modules should adhere to:

1. The module should be a subclass of `nn.Module`.
2. The forward method of the module should be capable of accepting any number of positional arguments with shapes denoted by $(B, *)$, where $B$ stands for the batch size, and $*$ denotes any other dimensions.
3. All input tensors should be located on the same computational device and should share the same data type (dtype).
4. The method should return a single tensor with the shape $(B, *)$.

The TorchCache library has proven invaluable in our experiments by allowing us to efficiently utilize the outputs of our pre-trained vision transformers without incurring significant computational overhead.

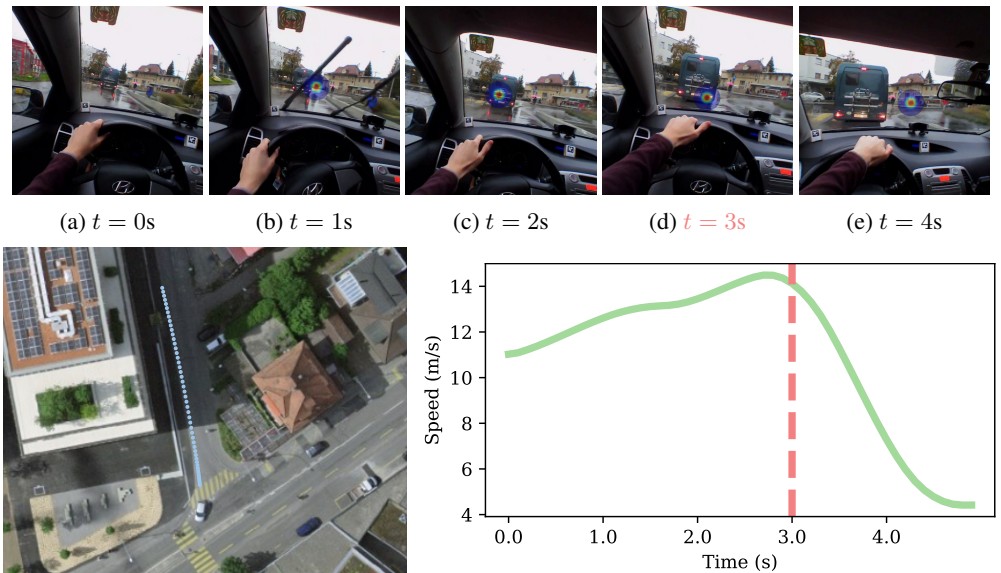

Figure 8: **A sample from the GEM dataset**. The top row shows frames with gaze heatmap overlay on top, at 1 Hz. The bottom left image displays the trajectory of the vehicle, while the bottom right plot shows the speed changes. This is an example of gaze-indicated short-term intent: the driver decelerates as soon as their gaze meets with the brake lights of the vehicle in front.

## B  EXPERIMENTS

### B.1  EVALUATION METRICS

The Average Displacement Error (ADE) is formally defined as:

$$\text{ADE} = \frac{1}{T} \sum_{t=1}^{T} \|\mathbf{p}_t^{\text{pred}} - \mathbf{p}_t^{\text{gt}}\|_2, \tag{1}$$

where $T$ is the total number of prediction time steps, $\mathbf{p}_t^{\text{pred}}$ is the predicted position at time $t$, and $\mathbf{p}_t^{\text{gt}}$ is the ground truth position at the same time.

The Final Displacement Error (FDE) is the displacement error at the final prediction time step:

$$\text{FDE} = \|\mathbf{p}_T^{\text{pred}} - \mathbf{p}_T^{\text{gt}}\|_2. \tag{2}$$

### B.2  HYPERPARAMETERS

The hyperparameters for the experiments can be found in Table 6, Table 7, Table 8, and Table 9.

### B.3  ANALYSIS OF FAILURE CASES

We observe two primary categories of prediction failures:

**1. Occlusions in right turns:** As shown in Figure 9a and 9b, trees and parked vehicles occlude crucial parts of the upcoming path during right turns. While human drivers naturally compensate for such occlusions through experience, our model shows increased uncertainty in these scenarios. Reliance on additional information from the car, such as turn signals or LIDAR, can improve driver behavior prediction for such instances.

**2. Complex intersections:** Figure 9c and 9d demonstrate varying accuracy in predicting motion at complex intersections and turns. The model particularly struggles determining when the driver is going to stop or resume in complex intersections (Figure 9c) and when multiple path options are available (Figure 9d). This limitation suggests an opportunity to incorporate past driver behavior to model driver initiative/aggression.

Table 6: Training configurations

| Parameter | Value |
| --- | --- |
| Optimizer | AdamW |
| Learning rate | 1e-5 |
| Weight decay | 1e-4 |
| Epochs | 200 |
| Epsilon | 1 |
| Visual epsilon | 0.3 |
| Scene video FPS | 1 |
| Gaze video FPS | 1 |
| Batch size | 16 |
| Output FPS | 5 |
| Loss discount factor | 0.97 |

Table 7: Framework configurations

| Parameter | Value |
| --- | --- |
| Encoder heads | 8 |
| Encoder layers | 8 |
| Feature dropout | 0.05 |
| Encoder hidden size | 64 |
| Gaze decoder layers | 2 |
| Gaze dropout | 0.2 |
| View dropout | 0.6 |
| Front video scaling factor | 0.3 |
| Scene video scaling factor | 0.1 |
| Gaze decoder dropout | 0.05 |
| Image embedding size | 64 |
| Dense visual loss ratio | 0.5 |

Table 8: GPS backbone (Informer) configurations

| Parameter | Value |
| --- | --- |
| Distil | true |
| Factor | 4 |
| Model dimension | 832 |
| Dropout | 0.0 |
| Number of Heads | 8 |
| Sequence Length | 40 |
| Decoder layers | 1 |
| Encoder layers | 6 |
| Prediction Length | 30 |
| Label Length | 40 |
| Activation | relu |
| Individual | false |
| Moving average | 25 |

Table 9: Video backbone (SwinV2) configuration

| Parameter | Value |
| --- | --- |
| Model Type | swinv2 window12to16 |
| Pad to square | true |
| Train backbone | false |

### B.4 ABLATION STUDIES ON MODALITY CONTRIBUTIONS

Following our architecture in Figure 2(a), we conducted ablation experiments by selectively removing modality branches from RouteFormer during inference. For FOV removal, we bypass the cross-modal transformer and use only scene features for self-attention. For scene removal, we retain only the FOV branch. When testing motion-only performance, we replace the entire visual feature tensor $v_{1:T}$ with zeros.

| Method | ADE (m) | Δ ADE (%) | ADE+20PCI (m) | Δ ADE+20PCI (%) |
| --- | --- | --- | --- | --- |
| **RouteFormer (full)** | 5.99 | Baseline | 7.70 | Baseline |
| motion+scene (w/o FOV) | 6.60 | +10.2 | 8.58 | +11.4 |
| motion+FOV (w/o scene) | 7.83 | +30.7 | 11.17 | +45.1 |
| motion only | 8.19 | +36.7 | 11.22 | +45.7 |

Table 10: Ablation study results showing degraded performance in removing different modalities.

The results, show in Tab. 10, demonstrate the clear benefits of each modality. Notably, our scene-only variant maintains performance consistent with RouteFormer-Base (6.13m), and our FOV-only variant remains competitive with GIMO (7.61m). This stability across modality configurations is particularly valuable for real-world deployment, where sensors or gaze tracking might be temporarily unavailable.

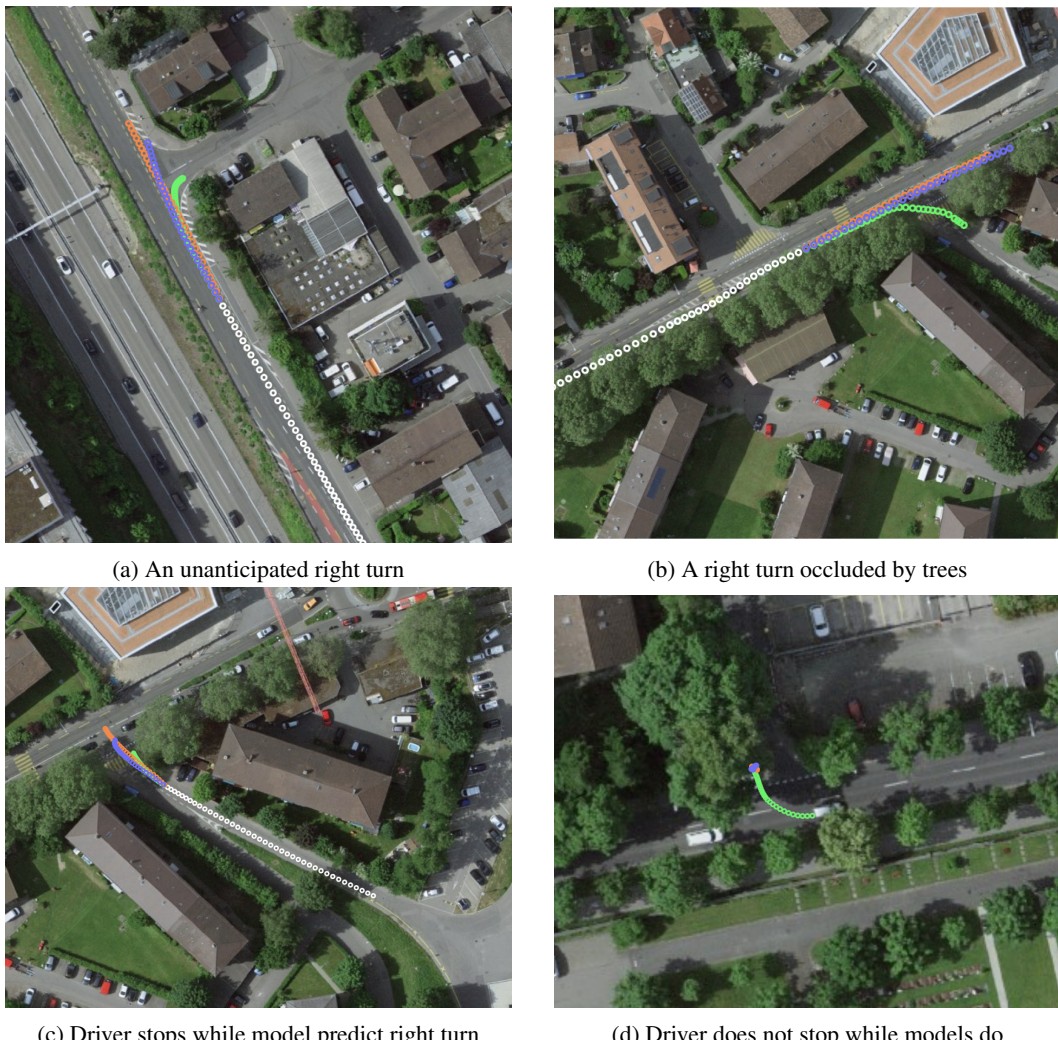

(a) An unanticipated right turn

(b) A right turn occluded by trees

(c) Driver stops while model predict right turn

(d) Driver does not stop while models do

Figure 9: **Failure cases analysis showing different scenarios where the model prediction deviates from ground truth (green).** Blue is RouteFormer, red is Routeformer-Base (without FOV modality), and white is the input trajectory.

### B.5 EXTENDED BASELINE SELECTION DISCUSSION

We detail here our approach to baseline selection and the challenges in adapting vehicle-specific models to our task. As the recent state-of-the-art methods work with multi-agent related traffic information, we consider such models (Autobots Girgis et al. (2021), MTR (Shi et al., 2022b)) as potential baselines.

We adapted Autobots Girgis et al. (2021) to our datasets by providing past ego-trajectory (ego_in) and setting other inputs (agents_in, road graph) to zeros/placeholders. The model's performance was significantly worse than our baselines (ADE of 10.87m on GEM). This underperformance stems from the fundamental mismatch between the input modalities and architectural design of these models, which are optimized for rich environmental context from multi-agent interactions rather than driver-centric prediction. The key architectural differences between ego-trajectory and multi-agent prediction approaches are summarized in Tab. 11.

Given the constraints, we focused on baselines that enable fair evaluation of our core contribution - integrating driver FOV data for trajectory prediction. GIMO and Multimodal Transformer

| Feature | MTR/Autobots | RouteFormer (Ours) |
|---|---|---|
| Primary Input | BEV, HD maps, LIDAR | Egocentric video, FOV data |
| Target Output | Multi-agent trajectories | Ego-vehicle trajectory |

Table 11: **Architectural differences between ego-motion and multi-agent prediction approaches.**

were selected because they support similar input modalities while representing the state-of-the-art in attention-based trajectory prediction.

## C    GEM DATASET

In this work, we proposed GEM, a multimodal ego-motion with gaze dataset with synchronized multi-cam video footage of the road scene, precise gaze locations from an eye-tracking headset and GPS ego-vehicle locations, manually corrected for further accuracy. We provide an additional sample scene from the dataset in Figure 8, demonstrating the connection between driver FOV and the short-term intentions of the driver.

### C.1    GROUND TRUTH CORRECTION WITH TRAJAPP

GPS data, particularly those collected using devices like GoPro cameras, can often be noisy and potentially inaccurate due to various reasons, ranging from satellite interference to obstructions like tall buildings or tunnels. Such inaccuracies in geospatial data can severely hamper the reliability and utility of a dataset, especially when the data is intended for detailed analysis and modeling. Examples of such cases, as well as milder issues, can be seen in Figure 10.

In response to this challenge, we have developed the desktop tool TRAJAPP , designed to facilitate the correction of noisy GPS data. The primary strength of TRAJAPP lies in its unique user interface which juxtaposes the video feed and the associated GPS location points on a map. This side-by-side presentation enables users to cross-verify the video content with the GPS coordinates visually.

Using the application is intuitive and straightforward, an example of which can be seen in Figure 11. As users watch the video playback, they can simultaneously observe the GPS path on the map. If discrepancies are noted, users can manually re-label the GPS data by simply clicking on the map and dragging the points to the desired accurate location. One key feature of TRAJAPP is its interpolation mechanism. Once users adjust certain key points on the map, the application automatically interpolates the GPS data between these points, ensuring smooth and consistent transitions.

Designed with user-friendliness in mind, TRAJAPP is cross-platform, with versions available for Windows, MacOS, and Linux. This ensures that researchers and dataset curators across diverse technological environments can access and utilize the tool to enhance the accuracy of their geospatial data. The application can be downloaded for free at github.com/meakbiyik/traj-app.

### C.2    GEM VS. DR(EYE)VE NOISE COMPARISON

To demonstrate the difference in degree of noise between the proposed GEM dataset and the driving gaze saliency dataset from the literature, DR(eye)VE, we plotted the distribution of the GPS distances to the nearest road center for each dataset in Figure 12. The road centers are gathered from the public data in OpenStreetMap. For GEM, the distance distribution is centered around 1.25 meters, which is consistent with the average lane width of 3 meters in urban settings, while the distribution is considerably more heavy-tailed in the case of DR(eye)VE, indicating significant GPS noise.

### C.3    GAZE FIXATION DETECTION

For our fixation detection, we adopted the methodology from Pupil Labs, the creators of the glasses embedded with gaze detectors (Tonsen et al., 2020). Our criteria define gaze fixations as instances

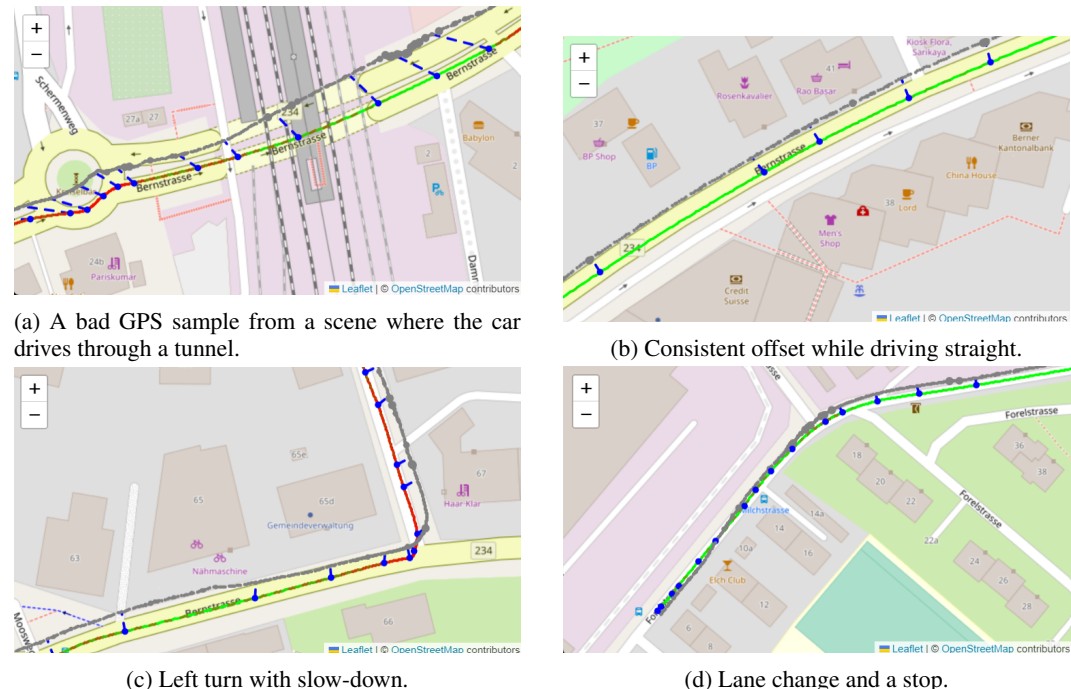

(a) A bad GPS sample from a scene where the car drives through a tunnel.

(b) Consistent offset while driving straight.

(c) Left turn with slow-down.

(d) Lane change and a stop.

Figure 10: **Example trajectory fixes with TRAJAPP .** The original GPS points are in gray, and the hand-placed markers are blue. The new markers are interpolated with a cubic spline with a color that represents the instantaneous speed: red for slow, and green for fast movement.

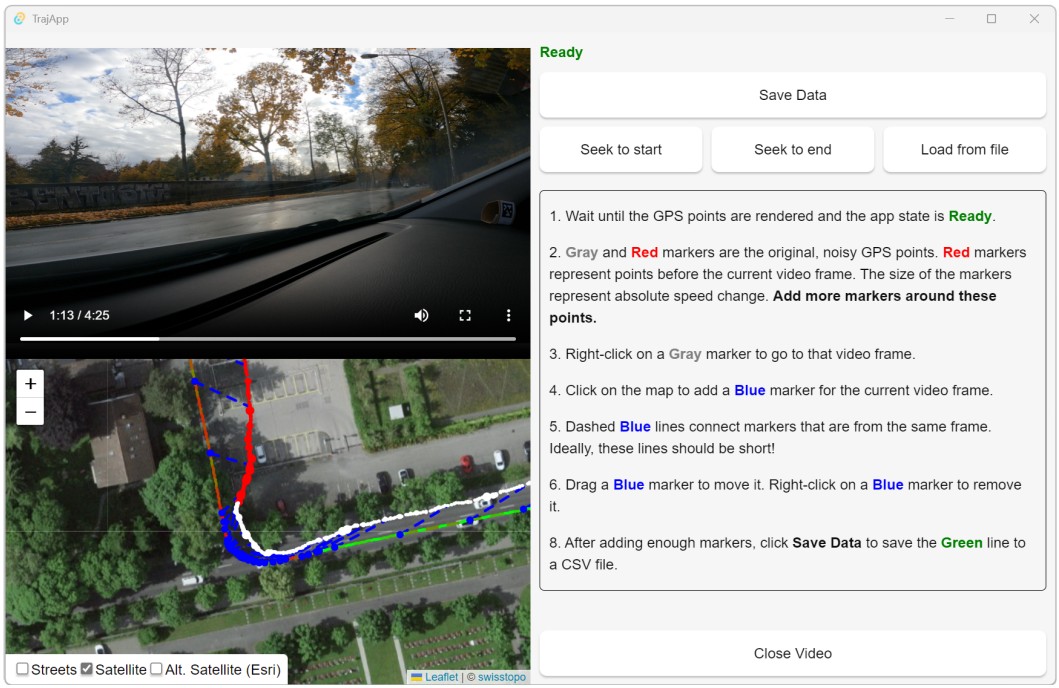

Figure 11: **User interface for GPS refinement.** We show a video and its corresponding GPS locations on a map concurrently. An annotator places correct markers on the map such that the GPS locations align with the video observations.

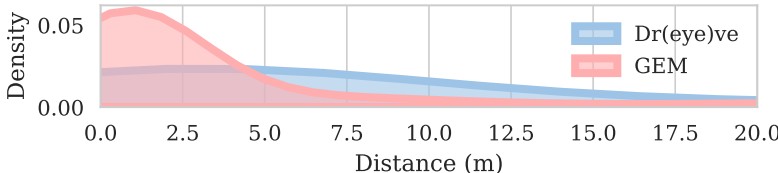

Figure 12: GPS distances to the nearest road center. Estimated via OpenStreetMap. DR(eye)VE's GPS points are further away from road centers, indicating significant noise.

where the gaze remains relatively stationary for a duration ranging from 80ms to 1 second. The upper limit of this range, 1 second, could also characterize a smooth pursuit, where the eyes smoothly track a moving object. Furthermore, we account for dispersion, which represents the spatial movement permitted within a single fixation. A gaze group is considered a fixation if the dispersion is within a limit of 1.5 degrees, ensuring that minor involuntary or random gaze movements do not undermine the detection of genuine fixations.

A set of example fixations can be seen in Figure 14. Note that there are frames without any detected fixations, which is explained by saccades and blinks. For others, the accuracy is high: note how the gaze meets exactly with the brake light of the bus in front, in Scene 1, Frame 9. Similarly, the gaze over the motorcycle in front is well-centered in Scene 5 for most frames.

## C.4 GAZE ALIGNMENT AND ROBUSTNESS

Our framework processes driver gaze data following established practices in eye-tracking research (Hayhoe et al., 2003; Land, 2006). To align the high-frequency gaze measurements (200 Hz) with video frames (5 Hz), we use median downsampling. This approach ensures robust position estimates while preserving the temporal characteristics of driver attention patterns.

To evaluate our model's robustness to potential measurement errors, we conducted additional experiments on the GEM dataset. As shown in Table 12, artificially introducing gaze position noise of $\pm50$ pixels (approximately 5% error) results in minimal degradation of prediction accuracy (0.01m increase in ADE). This resilience can be attributed to our "field-of-view" approach, which treats gaze as supplementary to the driver's head-mounted video feed rather than as a primary input.

| Configuration | ADE (m) | ADE+20PCI (m) |
|---|---|---|
| RouteFormer - without noise | 5.99 | 7.70 |
| RouteFormer - with $\pm50$px noise | 6.00 | 7.72 |
| RouteFormer-Base (without FOV) | 6.13 | 8.14 |

Table 12: **Impact of gaze measurement noise on prediction accuracy.** The minimal change in ADE demonstrates our model's robustness to gaze uncertainty. Top and bottom rows are from Table 2

These results demonstrate that our processing pipeline, built on standard gaze analysis techniques, effectively handles both natural gaze patterns and potential measurement uncertainties.

## C.5 EXPLORING IMAGE HOMOGRAPHY AND STITCHING

In our dataset, we provide synchronized videos from different viewpoints. In an attempt to provide a fully registered view from multiple cameras to our model, we have explored approaches for stitching them. However, we have discovered that using the separate videos brings benefits:

- Using the current state of the art, it is not feasible to register the driver headcam and the scene videos within an acceptable error margin. We experimented with classical feature extractors such as DISK (Tyszkiewicz et al., 2020) and transformer-based detector-free local feature matching (Sun et al., 2021), using Kornia library (Riba et al., 2019).

- We have observed that the redundancy from having multiple scene camera feeds can be exploited for training regularization. By dropping out one of the views at each frame during training, we improved the validation performance.

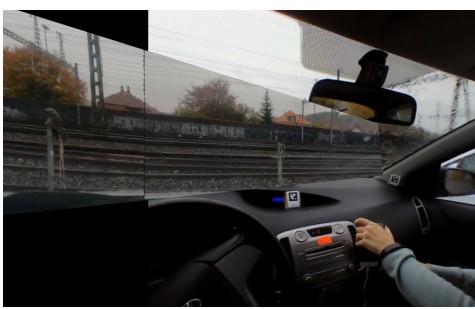 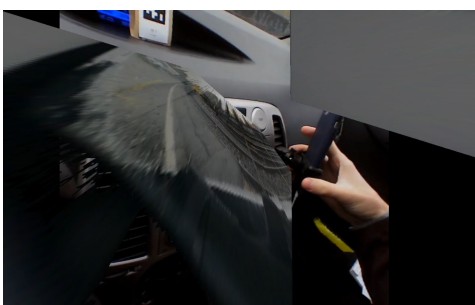

Figure 13: **Successful and failed stitching examples**. We used LoFTR for feature-matching across scene videos and the head-cam.

Example frames from the stitching experiments can be seen in Figure 13.

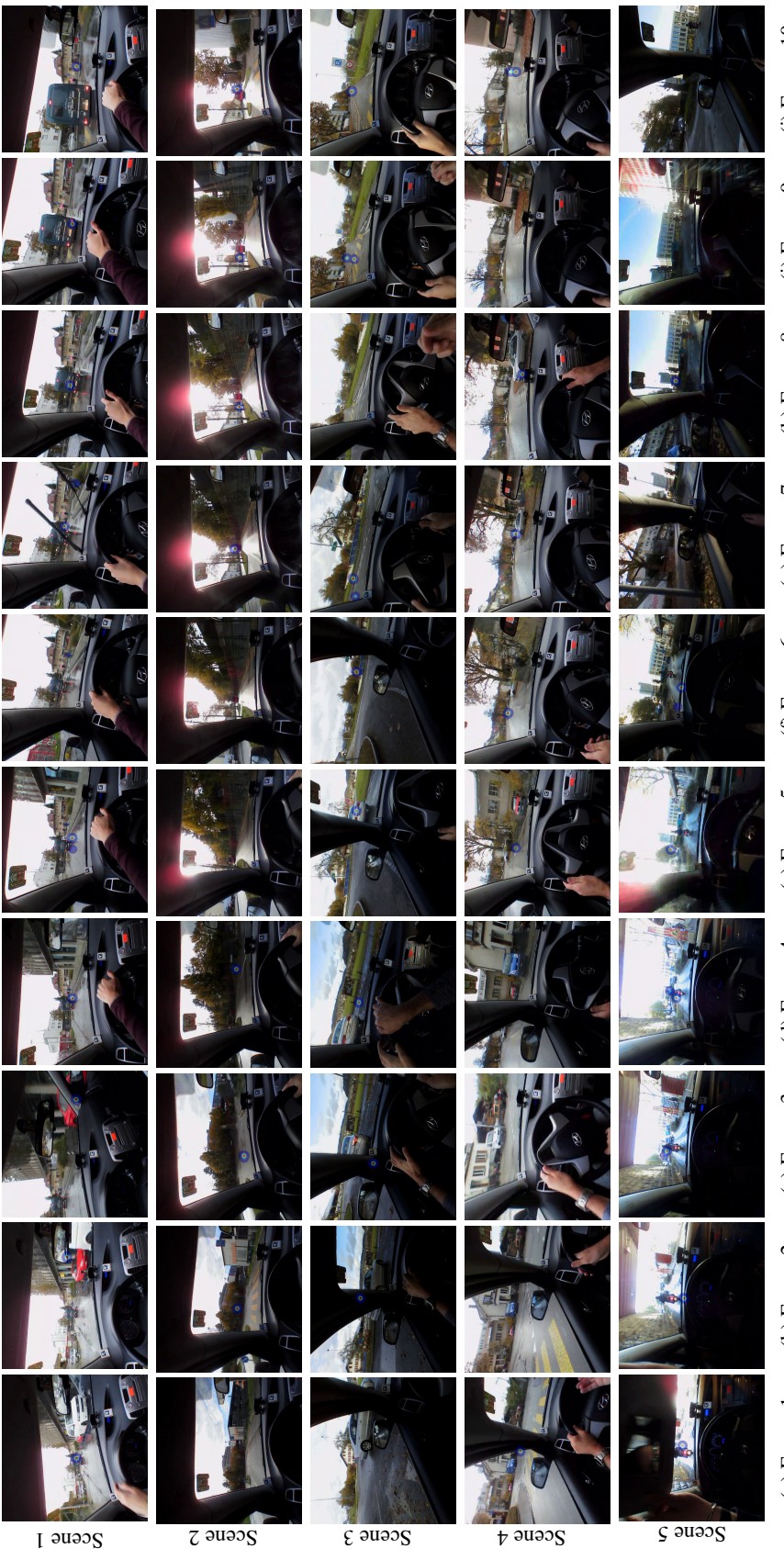

Figure 14: **Visual representation of gaze heatmaps across different scenes in the dataset.** Each scene row is visualized with a sample of 10 frames with a one-second time difference, illustrating gaze patterns over time. Note that fixations are not always available due to blinks and saccades.

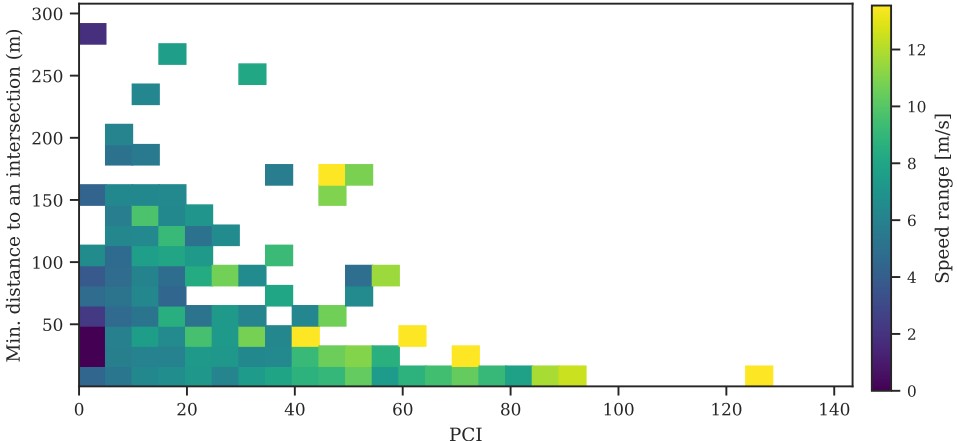

Figure 15: **Function of PCI with respect to the distance to an intersection.** Speed range–the difference between the highest and lowest instantaneous speed throughout the trajectory–is chosen instead of speed as high yet constant speed without a change in direction (e.g. cruising on a highway) is a low-PCI event. Bins with multiple samples are displayed, with a median speed range per bin. PCI is highly correlated with distance to an intersection, but only if a vehicle changes its speed throughout the trajectory, which discards stoppings and straight driving cases.

Table 13: **Breakdown of our GEM dataset.** We report duration, weather condition, and PCI for each participant.

| Subject | Total Distance (m) | Rainy | PCI (mean) | PCI (max) | Split |
|---------|-------------------|-------|------------|-----------|-------|
| 001 | 13,112 | ✓ | 12.37 | 92.99 | train |
| 002 | 12,824 | ✓ | 10.76 | 81.22 | val. |
| 003 | 12,997 | ✗ | 20.09 | 94.87 | train |
| 004 | 13,064 | ✗ | 14.59 | 79.30 | val. |
| 005 | 12,872 | ✗ | 17.29 | 86.39 | train |
| 006 | 13,127 | ✗ | 13.82 | 88.00 | train |
| 007 | 12,973 | ✗ | 16.42 | 77.47 | train |
| 008 | 13,700 | ✓ | 17.02 | 89.48 | test |
| 009 | 12,958 | ✗ | 15.30 | 103.96 | test |
| 010 | 13,048 | ✗ | 14.63 | 81.42 | train |

## D  PCI METRIC

### D.1  BREAKDOWN OF GEM DATASET AND PCI INSIGHTS

For the information on the dataset split and the behavior of each participant, see Table 13.

A deeper dive into our dataset, combined with synchronization with OpenStreetMap (OpenStreetMap contributors, 2017), reveals that the high PCI events indeed happen either close to the intersections, or on trajectories with a high-speed change, as observed in Figure 15.

### D.2  VISUALIZING PATH COMPLEXITY INDEX

Figure 16 illustrates the key components of PCI computation. Given an input trajectory (gray), we extrapolate a simple trajectory (dashed) by maintaining the final velocity vector. PCI measures the Fréchet distance (red) in meters between this simple trajectory and the actual target path (blue), effectively quantifying how much the actual path deviates from the simple projection. This deviation captures the complexity of the maneuver - higher deviations indicate more complex trajectories, requiring additional context to predict accurately. Unlike an alternative metric like mean-squared

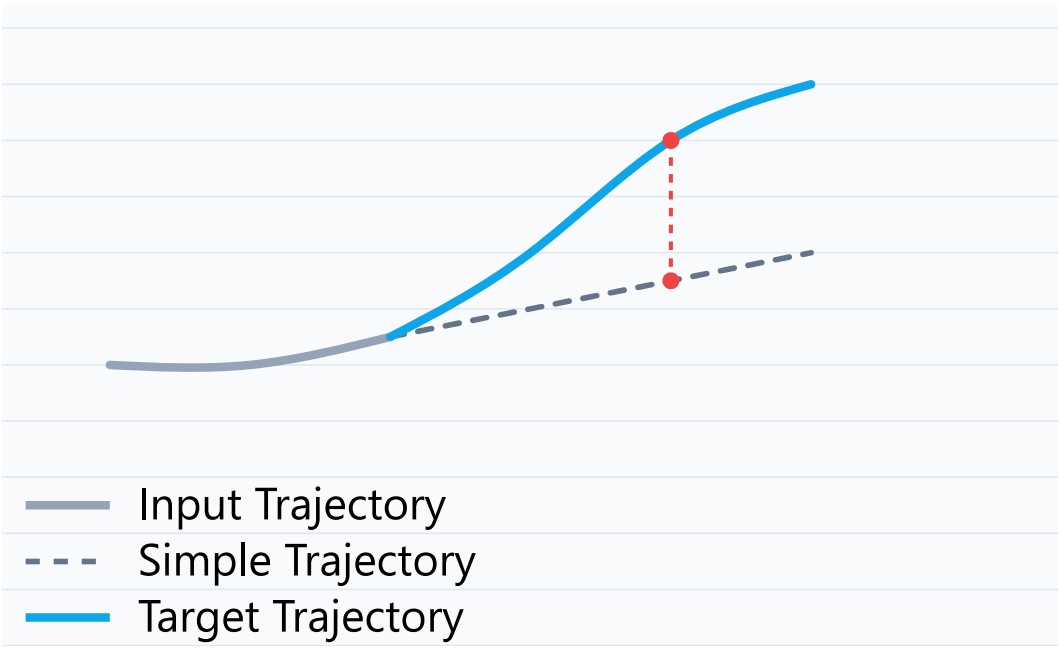

Figure 16: **A simple representation of PCI as a concept.**

error, Fréchet distance is more stable against speed changes, better capturing differences in trajectory rather than point-by-point error.

### D.3 PCI ANALYSIS ON URBAN DRIVING

Figure 17 visualizes PCI values across a representative urban drive in GEM. High PCI regions (bright yellow) consistently align with challenging scenarios: roundabout exits (top-right, PCI > 80), lane merges after intersections (bottom-left, PCI > 60), and sharp turns exceeding 45 degrees (top-left and bottom-right). Notably, slower 90-degree turns exhibit lower PCI values due to reduced divergence from simple trajectories, demonstrating PCI's ability to distinguish the degree of challenge to predict each target curve.

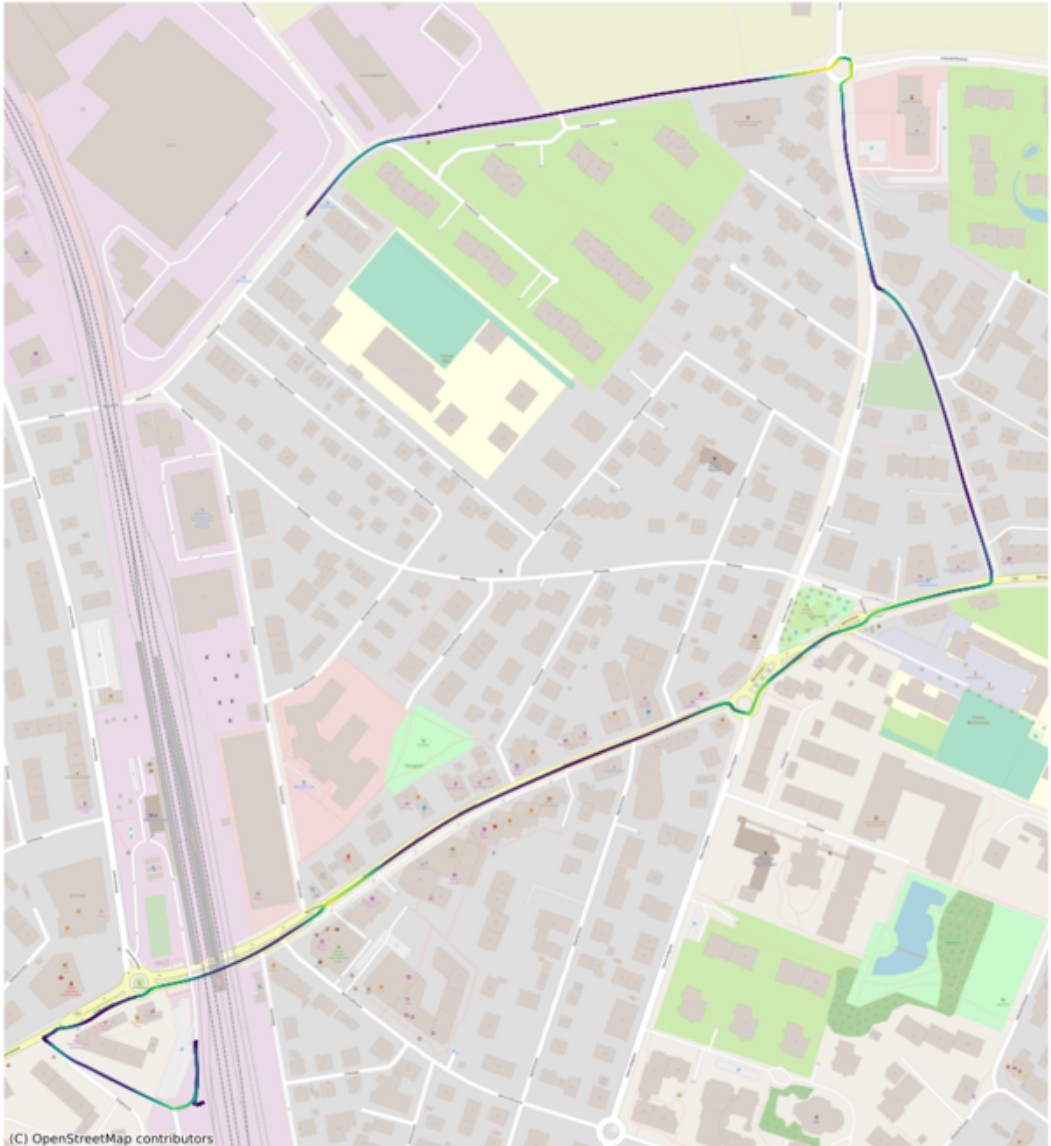

Figure 17: **Average PCI values of a full trajectory of a video in GEM.** Estimated by running a sliding window of 1 second across the whole trajectory with 6s-8s input-target length, and averaging the PCI values assigned to each point. Brightest yellow parts map to $\sim 80$ PCI while the dark blue is at 0 PCI.

