# OpenReview forum: "Leveraging Driver Field-of-View for Multimodal Ego-Trajectory Prediction"
_ICLR.cc/2025/Conference — ICLR 2025 Poster_

### Official Review · Reviewer_H88J · 2024-10-31

**Soundness:** 3
**Presentation:** 3
**Contribution:** 3
**Rating:** 6
**Confidence:** 4

**Summary:**

In this paper, the authors propose an end-to-end multimodal ego-motion prediction network that utilizes driver’s field-of-view data. Specifically, the proposed model predicts egocentric trajectory by taking scene videos, driver field-of-view videos, driver gaze position, and past trajectory as its input. Besides, the authors also propose a metric to measure the trajectory complexity, and an ego-motion dataset with driver positions and perspective. The experimental results show some improvements in the task of trajectory prediction task.

**Strengths:**

1. The proposed solution sounds solid in theory. Specially, considering driver’s attention and intent as well as surrounding environment is convincing. The driver’s attention and intent is a key impact factor of future trajectory.
2. The proposed dataset is helpful for other researchers to do more related research on that.
3. The proposed metric of trajectory complexity is helpful. It is a good indicator while analyzing the performance of a self-driving algorithm.
4. The ablation study and supplementary material are helpful. Readers can get more information from the ablation study results.

**Weaknesses:**

1. It will be better if the authors could report some failure cases. Especially for those cases which the predictions are totally opposite of the ground truth.
2. It will be better if the authors could compare their solution with latest schemes published in Year 2023 or 2024.

**Questions:**

1. What if the estimated gaze positions move quickly (either caused by wrong prediction or the driver moves his/her gazes quickly)? Is there any negative impact on the final results?
2. Line 190, the gaze positions are in the shape of Tx2. Does it mean we only use one gaze for each time stamp?

---

> ### Author Response · Authors · 2024-11-24
>
> Dear Reviewer H88J,
>
> We sincerely appreciate your thoughtful review and constructive feedback. We are particularly encouraged by your recognition of our solution's theoretical soundness and the value of our contributions to the dataset and metrics. We address your questions and concerns below:
>
> 1. **Failure Case Analysis**
>
> We have now added a new Section D.2 to the Appendix analyzing failure modes, with particular focus on cases where predictions significantly diverge from ground truth. We observe two primary failure patterns:
> - Right turns with partial occlusions by trees or buildings, and
> - Movements before/after a stop sign, due to individual driver behavior patterns
>
> These cases, particularly the stop sign scenario, highlight an interesting future direction: incorporating driver behavior modeling into trajectory prediction. We've included visualizations of these cases in the new Figure 17, along with a discussion of potential mitigation strategies.
>
> 2. **Comparison with Recent Work**
>
> Thank you for this suggestion. We have extended our related work to include recent works (Lines 102-107, Lines 127-132) and summarize the changes here:
>
> - Other recent related works we found for maneuver predictions, based on inferring driver’s intent: CEMFormer, employing in-cabin footage as a source of driver behavior; and HeadMon, using inertial head motion sensor data as intent-related data. We note that our task, trajectory prediction, is more advanced than maneuver prediction and that we focus on exploring the new driver’s field of view modality in the form of visual features to improve the prediction accuracy.
> - An overview of many other recent works that makes use of different modalities.
>
> The recent advances in trajectory prediction have primarily focused on multi-agent scenarios. In contrast, our work addresses the distinct challenge of ego-trajectory prediction using driver-centric data available in GEM and DR(eye)VE. Even the most recent work in this space, such as Ma et al.'s CEMFormer, uses in-cabin footage for intent classification rather than continuous trajectory prediction.
>
> Our attempts to adapt state-of-the-art vehicle trajectory predictors like Autobots to our setting (by providing only ego-trajectory while zero-padding other inputs such as map data) resulted in significantly worse performance (10.87m ADE on GEM), reinforcing that our modality-specific approach and carefully chosen baselines remain most appropriate in 2024 for evaluating driver-aware ego-trajectory prediction.
>
> 3. **Handling Rapid Gaze Movements and Measurement Errors**
>
> Human gaze naturally alternates between fixations (where gaze remains relatively stable on a point of interest) and saccades (rapid jumps between fixations). This fundamental pattern of human visual attention is well-documented in driving research (Land & Lee, 1994; Hayhoe et al., 2003). Following established practices in gaze processing (Pupil Labs, 2024), we handle these natural gaze patterns through:
> - **Standard fixation detection** as described in Appendix C.1, which identifies periods where gaze remains within a small spatial region (typically 80ms to 1 second), filtering out saccades and blinks, and
> - **Temporal alignment** through median downsampling, matching our high-frequency gaze data (200 Hz) to video timestamps (5 Hz) while ensuring robust position estimates.
>
> Importantly, our "field-of-view" approach treats gaze as supplementary to the driver's head-mounted video feed, providing robustness to the inherent discreteness of natural gaze patterns. As such natural patterns are already present in our training data, based on the results in Table 2, they are well-handled by our methodology.
>
> To quantify the robustness of our model to hardware-sourced gaze measurement errors, we conducted additional experiments on GEM, reporting the results in the table below. When we artificially introduce gaze position noise of ±50 pixels (~5% error), prediction accuracy only degrades by 0.01m ADE (0.02m on ADE+20PCI), demonstrating that our processing pipeline, built on standard gaze analysis techniques, effectively handles both natural gaze patterns and potential measurement uncertainties. We add this table to the Appendix in Section C.2 (the top and bottom rows were quoted from Table 2).
>
> | Configuration | ADE (m) | ADE+20PCI (m) |
> |-|-|-|
> | RouteFormer | 5.99 | 7.70 |
> | RouteFormer - with ±50px noise | 6.00 | 7.72 |
> | RouteFormer-Base (without FOV) | 6.13 | 8.14 |
>
> 4. **Gaze Position Representation**
>
> To clarify Line 190: Yes, we use one gaze position per timestamp. We downsample the approximately 40 gaze samples per video frame to one tuple per frame using fixation detection and median filtering. The Tx2 shape represents this final aligned data format, where each gaze position is represented as (x,y) coordinates matched to the corresponding video frame.
>
> We appreciate your feedback and believe addressing these points has helped clarify important aspects of our work.

---

> > ### Comment · Reviewer_H88J · 2024-11-25
> >
> > Thank you for your responses. The authors responded to my questions in a great way. They addressed my concerns. I do not have additional questions. Thank you so much.

---

> > > ### Author Response · Authors · 2024-11-26
> > >
> > > Dear Reviewer H88J,
> > >
> > > Thank you for confirming that our responses addressed your concerns. We are pleased that our additions - the failure analysis, recent comparisons, and robustness studies - have enhanced the manuscript. Given how these improvements align with your initial positive assessment, we kindly invite you to consider updating your rating. We remain committed to further strengthening the work for the camera-ready version.
> > >
> > > Sincerely,
> > > The Authors

---

### Official Review · Reviewer_15rW · 2024-11-02

**Soundness:** 4
**Presentation:** 3
**Contribution:** 3
**Rating:** 8
**Confidence:** 5

**Summary:**

The paper introduces RouteFormer, a multimodal ego-motion prediction network that integrates driver field-of-view (FOV) data with scene and GPS information to improve trajectory prediction. RouteFormer is the best solution on the market for this problem. Its Path Complexity Index (PCI) metric assesses trajectory difficulty and enhances model evaluation in complex scenarios. Additionally, the paper presents GEM, a new dataset containing synchronised driver FOV, gaze, and GPS data, capturing diverse urban driving conditions. RouteFormer significantly improves prediction accuracy, especially in high-complexity scenarios, advancing safety in driver-assistance systems.

**Strengths:**

This paper has done an impressively solid and hardcore job—it's a pleasure to read, providing a refreshing clarity that’s truly satisfying. It offers genuinely unique insights and, in my opinion, fully meets the standards for conference publication.

1. RouteFormer effectively combines driver FOV, scene, and GPS data, resulting in enhanced accuracy for predicting complex, non-linear trajectories. This multimodal integration marks a notable advancement over existing models.

2. The introduction of the Path Complexity Index (PCI) provides a novel way to quantify trajectory difficulty, allowing for better evaluation of model performance in challenging scenarios, which is often overlooked in traditional metrics.
The GEM dataset fills a significant gap by providing synchronized driver gaze, field-of-view, and GPS data, particularly suited for urban driving conditions with various traffic elements. This dataset enables more nuanced model training and testing, contributing to the broader research community.

3. Extensive testing demonstrates that RouteFormer performs robustly in real-world scenarios, highlighting its potential for practical applications in driver-assistance and AD systems.

**Weaknesses:**

Please refer to the Questions section.

**Questions:**

1. I have a couple of concerns regarding the definition of corner-case scenarios. The authors propose a model that essentially employs a contrastive learning approach—predicting a trajectory and then categorizing scenarios by comparing it with the ground truth. I believe this method introduces bias, as predictions differ across models. What evidence suggests that the prediction model used here is unbiased? While many recent works adopt similar methods to address long-tail issues, in this particular field, I find the approach still somewhat subjective.

2. The authors have not clearly defined what constitutes the long-tail phenomenon. It’s unclear why this method is capable of defining rare scenarios. I believe that this method, as described, does not adequately capture or define the essence of long-tail events. Could the authors provide further clarification?

3. The manuscript is inspired by human visual attention during driving, so it should cite earlier works that introduced similar ideas in trajectory prediction. These are:
(1) Human Observation-Inspired Trajectory Prediction for Autonomous Driving in Mixed-Autonomy Traffic Environments, IEEE International Conference on Robotics and Automation (ICRA 2024)
(2) Less is More: Efficient Brain-Inspired Learning for Autonomous Driving Trajectory Prediction, European Conference on Artificial Intelligence (ECAI 2024)

4. The related work section could benefit from a broader discussion of recent research. Currently, I find the discussion of the latest studies to be rather limited.

5. The paper would benefit from a discussion section that addresses its limitations and future challenges. This would provide a more comprehensive perspective.

6. Could the authors consider open-sourcing the project code and dataset? This would significantly contribute to the community.

Overall, I believe this paper meets the publication standards for ICLR.

---

> ### Author Response · Authors · 2024-11-24
>
> Dear Reviewer 15rW,
>
> We appreciate your thorough and insightful review. Your recognition of our paper's technical depth and clarity is particularly meaningful, as we invested significant effort in presenting the complex multimodal architecture clearly. We are encouraged by your positive assessment of our key contributions - the GEM dataset, RouteFormer architecture, and PCI metric. We address your questions below.
>
> 1. **PCI Definition and Bias**
>
> We appreciate your important question about potential bias in identifying corner cases, for which we use the novel PCI metric as an indicator.
>
> As a metric of complexity, PCI compares the target ground truth trajectory $T_{target}$ with an imaginary linear trajectory $T_{simple}$, where the vehicle maintains the final speed and angle from the input sequence. A newly added Figure 10 to Appendix A.2 illustrates this "simple" trajectory in an example.
>
> Consequently, the model predictions themselves are not used to evaluate trajectories, nor as part of a contrastive loss scheme. Rather, we use a basic extrapolation (which is also our linear baseline in Table 2) to categorize all trajectories. Through its simplicity and consistency, this design avoids introducing model bias to the evaluation phase.
>
> PCI is crucial for our analysis: in Figures 6 and 7 it reveals that gaze's impact is very significant in the hard samples. Similarly, in Table 2, it allows us to easily filter trivial trajectories with <20 PCI, emphasizing improvements achieved via different modalities. Finally, it correlates with our expectations of long-tail trajectories, as we explore below.
>
> 2. **Long-Tail Phenomenon**
>
> Consistently with both GEM and DR(eye)VE datasets, around 75% of trajectories have less than 20 PCI. Referring to Figure 4, we consider these instances as cases of straight cruising with little to no speed change. However, you rightly ask about the remaining 25% of samples.
>
> While driver intent prediction often relies on hand-labeled events (as in the LOKI dataset from Girase et al.), with turns, lane changes and "cut-ins" requiring human consensus on the action, we take a different approach. We consider rare scenarios not as hand-labeled fixed targets, but as trajectories with unexpected continuations given initial behavior: drastic angle or speed changes, and non-linear paths.
>
> This seemingly simple benchmark aligns surprisingly well with driver-perceived challenges. Our newly added Figure 11 in the Appendix A.3 employs a sliding window to assess average PCI across a GEM dataset video. In this sample, the following situations register as high PCI:
>
> - **Roundabouts:** highest PCIs are observed in roundabouts, with using the third exit registering higher PCI than using 1st or 2nd exits.
> - **Lane merges:** high PCI is recorded due to a slowdown and a change of direction
> - **Turns:** turn angle appears to be an important factor that elevates PCI, together with the turn speed
>
> Despite its simplicity, PCI powerfully detects relevant driving events without requiring map metadata, demonstrating our model's strengths.
>
> 3. **Related Work Improvements**
>
> Thank you for noting the relevant ICRA and ECAI 2024 papers. We've updated our manuscript and incorporated these citations in our related work. In addition, we expand our related work discussion (Lines 102-107, Lines 127-132) and summarize the changes here:
>
> - The addition of human-inspired trajectory prediction works as recommended by the reviewer,
> - Recent maneuver prediction works using inferred driver intent: CEMFormer, which explores the benefit of in-cabin driver behavior monitoring for maneuver prediction; and HeadMon, which uses inertial head motion data for maneuver prediction. In contrast, we perform the more complex trajectory prediction task and employ driver’s visual field of view,
> - A discussion on a few recent works in the multi-agent version of trajectory prediction using different modalities.
>
> The literature remains near-exclusively multi-agent, and our paper preserves its edge for ego-trajectory prediction together with its unique addition of driver field-of-view.
>
> 4. **Additional Discussion**
>
> Thank you for this valuable suggestion. We have added a new Section D.2 to the Appendix analyzing key failure modes in the GEM dataset, particularly cases involving occluded right turns and varying driver behaviors at stop signs. These examples highlight both current limitations and promising research directions in driver behavior modeling.
>
> 5. **Open-Sourcing the Project Code and Dataset**
>
> We confirm that both the complete codebase and GEM dataset will be made publicly available upon publication, including:
>
> - Full dataset (GEM) with the ground truth
> - Model implementation and training code
> - Data collection and preprocessing pipeline
> - PCI computation tools
> - GPS correction tooling
>
> We thank you again for your constructive feedback that has helped significantly strengthen both the clarity and technical depth of our manuscript.

---

> > ### Comment · Reviewer_15rW · 2024-11-25
> >
> > Thank you for your reply! I think the authors have addressed my concerns very well. I really appreciate and am encouraged by the authors' dedication and practical approach. I'll improve my rating to help your work gain more recognition and serve as a valuable reference for others. Thank you for your contributions to the AD community - this is excellent work!

---

> ### Author Response · Authors · 2024-11-26
>
> Dear Reviewer 15rW,
>
> We are grateful for your thorough review and valuable feedback, which has significantly enhanced our manuscript. Your recognition of our work's contribution to the AD community is particularly meaningful, and we look forward to supporting future research through our open-sourced code and dataset.
>
> Sincerely,
>
> The Authors

---

### Official Review · Reviewer_U1qP · 2024-11-04

**Soundness:** 3
**Presentation:** 4
**Contribution:** 3
**Rating:** 6
**Confidence:** 5

**Summary:**

The paper introduces a multimodal ego-trajectory prediction model that utilizes driver field-of-view (FOV) data—including first-person video and gaze fixations—along with environmental and GPS information to enhance the accuracy of vehicle trajectory forecasting. The method surpasses conventional models through the implementation of a Path Complexity Index (PCI), which assesses the complexity of diverse driving scenarios. The authors present a novel dataset that provides comprehensive urban driving data enriched with driver FOV information.

**Strengths:**

- New dataset: the authors contribute GEM, a high-quality dataset focused on urban driving, which includes synchronized FOV and gaze data, filling a critical gap in available resources for trajectory prediction. It will be good to have this dataset for the autonomous driving community.
- The proposed RouteFormer introduces a unique approach by integrating driver field-of-view (FOV) data, such as first-person video and gaze fixations, with GPS and environmental data, enhancing the accuracy of ego-trajectory prediction.
- Introduction of Path Complexity Index (PCI). The paper proposes a novel PCI metric to quantify the complexity of driving scenarios, offering a more nuanced assessment of prediction challenges compared to traditional metrics.
- The proposed method outperforms the other baselines on both datasets.

**Weaknesses:**

- To fully understand how much each modality contribute to the improvement, it will be interesting to add the ablations studies below and compare the ADE and ADE+20PCI.
   1. RouteFormer without video and gaze (Motion only).
   2. RouteFormer without surrounding scenes (Motion + Gaze).

- Stronger baselines are needed. The current baselines (GIMO, Multimodal Transformer) used in this paper are mainly designed for human motion. Compared with at least one vehicle motion predictor is more convincing (e.g., MTR [1], Autobots [2], ...). Or are there any potential reasons that the authors thought the human motion baselines were more appropriate than vehicle baselines?

[1] Shi, Shaoshuai, et al. "Motion transformer with global intention localization and local movement refinement." Advances in Neural Information Processing Systems 35 (2022): 6531-6543.
[2] Girgis, Roger, et al. "Latent variable sequential set transformers for joint multi-agent motion prediction." International Conference on Learning Representations (ICLR), 2022.

**Questions:**

Please refer to the weaknesses above.

---

> ### Author Response · Authors · 2024-11-24
>
> Dear Reviewer U1qP,
>
> Thank you for your thoughtful review and valuable feedback on our paper. We appreciate your recognition of our contributions, and have conducted additional analyses to address your concerns comprehensively, which have led us to interesting findings.
>
> 1. **Ablation Studies on Modality Contributions**
>
> Following our architecture in Figure 2(a), we conducted ablation experiments by selectively removing modality branches from RouteFormer (full, trained model) during inference. For FOV removal, we bypass the cross-modal transformer and use only scene features for self-attention. Similarly, for scene removal, we retain only the FOV branch. To compare the motion-only performance with them, we replace the entire visual feature tensor $v_{1:T}$ with zeros. The results reveal several interesting findings about RouteFormer's behavior and robustness.
>
> | Method | ADE (m) | ADE+20PCI (m) |
> |--------|---------|-----------|
> | **RouteFormer (full model)** | 5.99 | 7.70 |
> | motion+scene (without FOV) | 6.60 |  8.58 |
> | motion+FOV (without scene) | 7.83 |  11.17 |
> | motion only | 8.19 | 11.22 |
>
> The results demonstrate clear benefits of each modality: removing FOV increases ADE by 10.2%, while removing scene information leads to a 30.7% increase in error.
>
> Notably, our scene-only variant maintains performance consistent with the RouteFormer-Base (6.13m), which was trained directly with these inputs, and our FOV-only variant remains competitive with GIMO (7.61m) as shown in Table 2. In addition, the motion-only variant still shows significant improvement over the linear baseline (13.37m) for ADE+20PCI range. This stability across modality configurations is particularly valuable for real-world deployment, where sensors or gaze tracking might be temporarily unavailable, allowing graceful performance degradation rather than catastrophic failure when inputs are missing.
>
> These findings have been added to a new ablation studies section of the Appendix D.3.
>
> 2. **Choice of Baseline Methods**
>
> We appreciate the suggestion to include vehicle-specific models like MTR [1] and Autobots [2] as baselines to strengthen our evaluation. We have carefully considered this and would like to provide a detailed explanation of our approach.
>
> We attempted to adapt Autobots to our datasets by using only the available inputs, and train it on GEM from scratch. Specifically, we provided the past ego-trajectory (`ego_in`) while setting trajectory of other agents (`agents_in`) and road graph information (`roads`) to zeros or placeholders, as expected by the model signature. However, the model's performance was significantly worse than our baselines, yielding an ADE of 10.87m on GEM.
>
> This is due to the native incompatibility of our task with the input modalities, as explained in the table below.
>
> | Feature | MTR/Autobots | RouteFormer (Ours) |
> |---------|--------------|-------------------|
> | Primary Input | BEV, HD maps, LiDAR (multi-agent detection) | Egocentric video, FOV data |
> | Target Output | Multi-agent trajectories | Ego-vehicle trajectory |
>
> Overall, multi-agent trajectory prediction models like Autobots and MTR are architecturally designed to leverage rich environmental context from multi-agent interactions. Ego-trajectory prediction, however, is an inherently different task that relies on the view and intention of the driver, often without the availability of additional metadata.
>
> Given these constraints, we focused on baselines that enable fair evaluation of our core contribution - integrating driver FOV data for trajectory prediction. GIMO and Multimodal Transformer were selected as they support similar input modalities while representing SOTA in attention-based trajectory prediction.
>
> In light of this question, we expanded our discussion of baseline selection criteria with a section D.4 in the Appendix, and include detailed comparisons of architectural differences between ego-motion and multi-agent prediction approaches.
> We thank you again for your constructive feedback that has helped strengthen our experimental validation.

---

### Meta-Review · Area_Chair_4nyP · 2024-12-22

**Metareview:**

The present work proposes a multimodal ego-motion prediction framework that considers a driver's field of view. It essentially has three contributions: the prediction network itself, a new metric for measuring trajectory complexity, and a dataset involving multicamera footage of the scene and synchronized gaze data. The main strengths recognized by the reviewers are the empirical performance, the usefulness of the dataset, and the novelty of the complexity metric. The main weaknesses of the work that were initially mentioned were a lack of ablation studies, insufficient recency of baselines, and several further clarification requests. Most of these concerns have been addressed and while the authors did not significantly expand experimental comparisons, they provided a sufficient justification. Thus, overall, the reviewers and I agree that this work is worthy of acceptance.

**Additional Comments On Reviewer Discussion:**

The authors addressed the questions and weaknesses raised by the reviewers and provided ablation information. As a consequence, one reviewer (15rW) stated that they increased the rating, and overall, all reviewers agree that this work is above the acceptance threshold.

---

### Decision · Program_Chairs · 2025-01-22

Accept (Poster)